# Neutrophil breaching of the blood vessel pericyte layer during diapedesis requires mast cell-derived IL-17A

Régis Joulia [1,6], Idaira María Guerrero-Fonseca [1,2], Tamara Girbl[1,7], Jonathon A. Coates [1], Monja Stein[1], Laura Vázquez-Martínez[1], Eleanor Lynam[1], James Whiteford [1], Michael Schnoor [2], David Voehringer [3], Axel Roers [4], Sussan Nourshargh[1,5,8] & Mathieu-Benoit Voisin [1,5,8] ✉

Neutrophil diapedesis is an immediate step following infections and injury and is driven by complex interactions between leukocytes and various components of the blood vessel wall. Here, we show that perivascular mast cells (MC) are key regulators of neutrophil behaviour within the sub-endothelial space of inflamed venules. Using confocal intravital microscopy, we observe directed abluminal neutrophil motility along pericyte processes towards perivascular MCs, a response that created neutrophil extravasation hotspots. Conversely, MC-deficiency and pharmacological or genetic blockade of IL-17A leads to impaired neutrophil sub-endothelial migration and breaching of the pericyte layer. Mechanistically, identifying MCs as a significant cellular source of IL-17A, we establish that MC-derived IL-17A regulates the enrichment of key effector molecules ICAM-1 and CXCL1 in nearby pericytes. Collectively, we identify a novel MC-IL-17A-pericyte axis as modulator of the final steps of neutrophil diapedesis, with potential translational implications for inflammatory disorders driven by increased neutrophil diapedesis.

Neutrophil migration through blood vessels is a vital component of innate immunity against infectious agents and sterile injury. To exit the blood circulation and enter inflamed tissues, neutrophils exhibit sequential cellular and molecular interactions with different components of the venular wall, namely endothelial cells (EC), pericytes and the venular basement membrane (BM)[1–3]. Whilst the events driving neutrophil-EC interactions have been extensively studied, deciphering the molecular basis of neutrophil migration within the sub-EC space and breaching of pericytes in vivo has received little attention[2,4,5]. In this context, previous studies have reported the expression of adhesion molecules (e.g., E-selectin, ICAM-1 and VCAM-1)[6,7], and chemokines (e.g., CXCL1, CXCL8, IL-6, MIF)[6,8] on human pericytes following in vitro stimulation. Furthermore, we and others demonstrated that pericytes expressed pro-inflammatory receptors (e.g., TNFR1/2, IL-1R, TLR4), adhesion molecules and chemokines following acute inflammation in vivo[9–11]. Moreover, intravital confocal imaging showed that neutrophils exhibit significant abluminal crawling on pericyte processes (~20 min) post breaching the endothelium, where after, they migrate through preferential exit sites within the pericyte layer to leave the venular wall[9,10]. However, the cellular and molecular basis of this final stage of neutrophil diapedesis remains unknown.

[1]William Harvey Research Institute, Faculty of Medicine and Dentistry, Queen Mary University of London, Charterhouse Square, London EC1M 6BQ, UK. [2]Department of Molecular Biomedicine, CINVESTAV-IPN, Mexico City, Mexico. [3]Department of Infection Biology, University Hospital Erlangen and Friedrich-Alexander University Erlangen-Nuremberg (FAU), Erlangen 91054, Germany. [4]Institute for Immunology, Heidelberg University Hospital, Heidelberg, Germany. [5]Centre for Inflammation and Therapeutic Innovation, Queen Mary University of London, London EC1M 6BQ, UK. [6]Present address: NHLI, Imperial College London, London, UK. [7]Present address: Rudolf Virchow Center for Experimental Biomedicine, University of Würzburg, Würzburg, Germany. [8]These authors contributed equally: Sussan Nourshargh, Mathieu-Benoit Voisin. ✉e-mail: m.b.voisin@qmul.ac.uk

Mast cells (MCs) are prototypical immune sentinels that reside in most peripheral tissues, often in close vicinity to microvessels[12–14]. As a prominent cellular source of a multitude of inflammatory (e.g., cytokines, chemokines) and vasoactive (e.g., histamine, leukotrienes, serotonin) mediators, MCs are considered to play a critical role in triggering the onset and development of acute inflammatory reactions[15]. As such, there is ample evidence for the ability of MCs to support neutrophil recruitment in vivo, including direct evidence using MC deficient mice in models of *K. pneumonae* infection, IgE or hapten-dependent cutaneous inflammation, and following LPS stimulation[13,16–18]. Despite these studies, the mechanisms of MC-dependent neutrophil migration through blood vessel walls remain unclear.

To address this unexplored element of acute inflammation, we hypothesise that perivascular MCs contribute to the programming of pericytes towards an adhesive phenotype as a mean to promote efficient and localised neutrophil diapedesis. In this study, we use confocal intravital microscopy to simultaneously track the movement of neutrophils through inflamed venular walls in relation to ECs, pericytes and perivascular MCs in real-time. We observe that once in the sub-EC space, neutrophils exhibit a preferential and directed abluminal motility within the pericyte sheath towards areas enriched in perivascular MCs; a response that is defective in MC-deficient mice. At the molecular level, the directed motility of neutrophils is associated with higher expressions of ICAM-1 and CXCL1 on pericytes in close proximity to MCs. This response is mediated by MC-derived interleukin-17A (IL-17A), identifying regulation of neutrophil breaching of the pericyte layer as a previously unknown role for this cytokine. Together, by detecting a cross-talk between perivascular MCs and pericytes, we report on a novel axis involving MC-derived IL-17A as a driver of the final steps of neutrophil diapedesis.

## Results

### Perivascular mast cells promote localised neutrophil extravasation

To explore the role of MCs in neutrophil migration through venular walls, we employed an in vivo model of acute inflammation as induced by local injection of TNF in the mouse cremaster muscle. Initial works using whole-mount immunostained fixed tissues revealed a heterogeneous distribution of MCs with perivascular regions being either enriched or completely devoid of MCs (Fig. 1a, Supplementary Fig, 1a). A similar heterogenous distribution of MCs along blood vessels was noted in the myocardium and hindlimb skeletal muscle (Supplementary Fig. 1a). In contrast MC density was higher and more evenly distributed in the ear skin, as supported by the literature[12–14]. Of note, in the cremaster muscle, MCs were predominantly associated with arterioles and post-capillary venules (PCVs) with very few being aligned with capillaries (Supplementary Fig. 1b). Most importantly, ~80% of perivascular MCs were localised in juxtaposition (i.e., <20 μm away from vessel walls) to PCVs of diameters between 20 and 40 μm (Supplementary Fig. 1c), vessels known to support the majority of neutrophil extravasation[19]. Comparatively, resident macrophages and dendritic cells were evenly distributed along the three types of blood vessels (Supplementary Fig. 1d–f), with no significant differences noted in their number in relation to different-sized PCVs (Supplementary Fig. 1g). Furthermore, acute inflammation as induced by locally administered TNF did not alter the number or perivascular distribution of MCs or macrophages as compared to control (PBS) tissues (Supplementary Fig. 1h).

In quantifying total tissue infiltration of neutrophils, we detected a clear association between sites of intense neutrophil extravasation and MC-enriched venular segments (Fig. 1a, b). Correlation analysis showed this to be significant in both TNF-stimulated cremaster muscles and mouse ear skin (Fig. 1b, c). Together, these findings suggest that perivascular MCs support localised neutrophil extravasation. To

directly investigate this notion, we developed a 4-colour confocal intravital microscopy (IVM) approach to simultaneously analyse the migratory dynamics of neutrophils through ECs and the pericyte layer in relation to MC localisation. This model employed the compound mouse reporter strain *LysM-EGFP-ki;α-SMA-RFPcherry-Tg* exhibiting endogenous GFP^high neutrophils and RFP⁺ pericytes and smooth muscle cells[9,10]. The mice also received a local injection of non-blocking fluorescently labelled anti-CD31 mAb[19] and anti-CD117 mAb, delineating EC junctions and MCs, respectively (Fig. 1d). The effectiveness of our in vivo antibody-based MC labelling strategy was confirmed using MC reporter mice (*Mcpt5-Cre YFP*)[20,21] that showed all endogenously fluorescent (YFP⁺) MCs were labelled with the anti-CD117 mAb (Supplementary Fig. 2). Using this confocal IVM platform, we noted a significant increase in adhesion and crawling of neutrophils on the luminal aspect of post-capillary venules from 2 h post local injection of TNF. Crucially, the method enabled us to distinguish the key steps of neutrophil diapedesis, namely transendothelial migration (TEM), abluminal (sub-EC) motility along pericyte processes and breaching of the pericyte layer (a step we have termed transpericyte migration; TPM) in relation to localisation of perivascular MCs (Fig. 1e). In terms of TEM, most neutrophils breached the EC barrier in a paracellular fashion in line with our previous work[19].

Surprisingly, only a minority of TEM events (~30%) occurred in close proximity (<20 μm) of a perivascular MC, suggesting that MC localisation did not influence sites of neutrophil TEM (Fig. 1f and Supplementary Movie 1). In contrast, more than two-thirds (~70%) of neutrophils exited the venular wall through the pericyte sheath at venular regions in close apposition (<20 μm) to perivascular MCs (Fig. 1f, Supplementary Movie 2). To explore this "post TEM" tropism towards MCs, we analysed the behaviour and motility of neutrophils in the sub-EC space. Here, neutrophils exhibited a notable directional motility towards venular wall exit sites located near (<10 μm away) a perivascular MC as compared to MC-free venular segments (>10 μm away; Fig. 1g and Supplementary movie 3). Quantitative analysis of migration patterns in the abluminal space showed that neutrophil paths were shorter (~42%), faster (~40%) and straighter (~76%) for neutrophils migrating towards MCs as compared to behaviour of neutrophils towards MC-devoid venular segments (>10 μm away, Fig. 1h). This migratory profile resulted in establishment of notable hotspots of neutrophil TPM in close apposition to perivascular MCs, commonly involving more than 3 consecutive neutrophils exiting the venular wall within the same area. On average we noted ~2 hotspots per 300 μm length venular segment, with ~70% of hotspots occurring in the vicinity of perivascular MCs (<10 μm) (Fig. 1i, j; Supplementary Movies 4 and 5) and representing ~50% of all neutrophil TPM events. Of note, <20% of hotspots were localised in close proximity to perivascular macrophages (Fig. 1j), despite showing a more abundant and homogenous distribution along venular walls (Supplementary Fig. 1d, f–h).

Together these data align perivascular MCs with efficient neutrophil sub-EC motility and breaching of venular walls.

### Neutrophil abluminal motility is defective in MC-deficient mice

To investigate a potential causal link between perivascular MCs and neutrophil diapedesis, we extended our analysis to mice exhibiting constitutive MC deficiency. Based on the reported localisation of perivascular connective-tissue MCs[22] and our observations of the absence of mMCP-1⁺ (MCPT1⁺) MCs in cremaster muscles of WT mice (Supplementary Fig. 3a), we used the specific connective-tissue MC-deficient mouse line *Mcpt5-Cre⁺-RDTA/RDTA (termed* MC^deficient mice). The application of confocal microscopy and flow cytometry confirmed that the cremaster muscles of MC^deficient mice were devoid of MCs (Supplementary Fig. 3b–d). Preliminary works using bright-field IVM showed no defects in rolling and firm adhesion of leukocytes within TNF-stimulated cremaster muscle venules of MC^deficient mice as

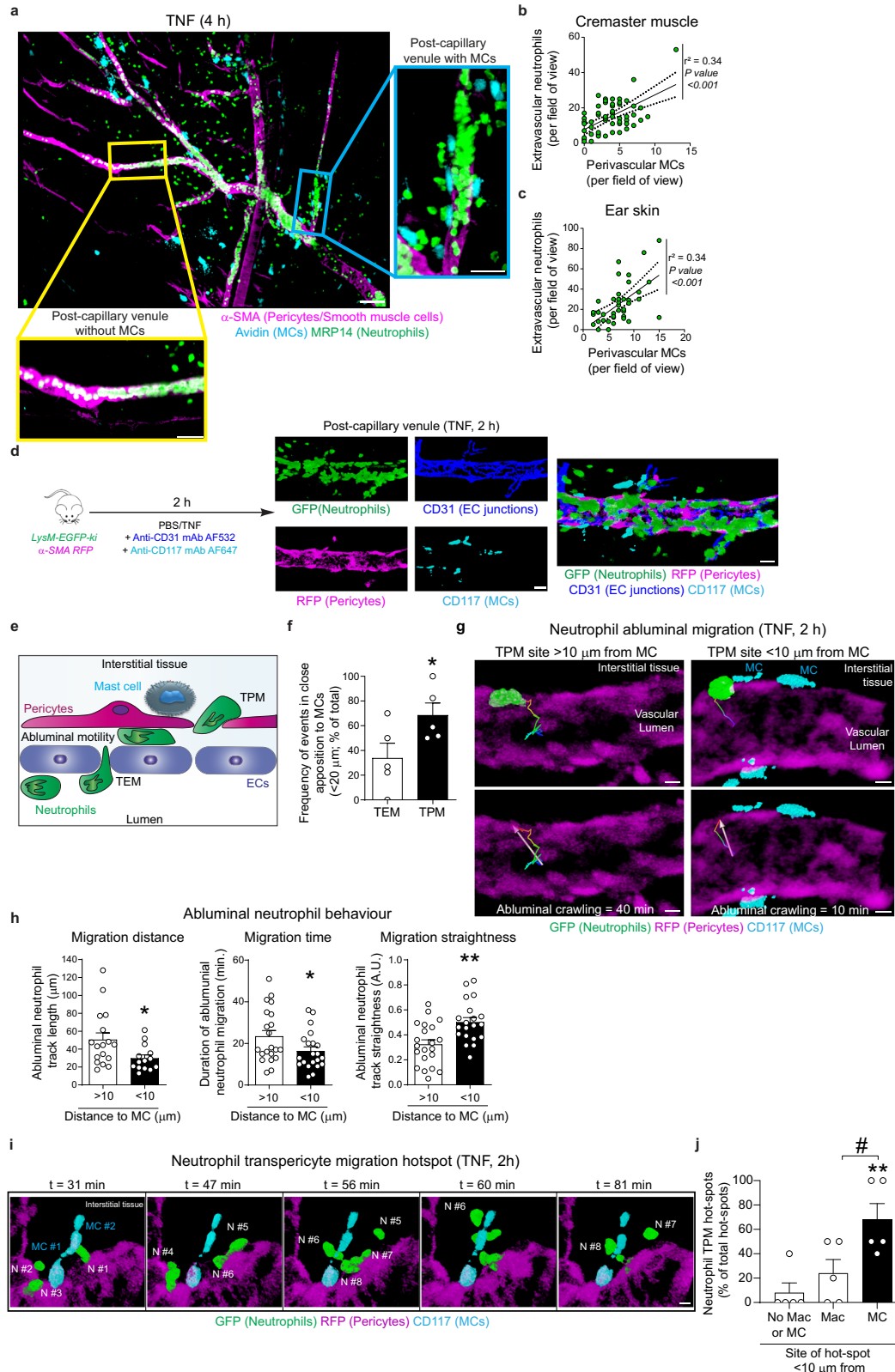

compared to littermate controls (Supplementary Fig. 3e, f), indicating MCs were not essential regulators of luminal neutrophil-EC interactions. To investigate the role of MCs in neutrophil TEM and TPM we crossed *Mcpt5-Cre⁺-RDTA* mice with *LysM-EGFP-ki;α-SMA-RFPcherry-Tg* animals, using the littermates *Mcpt5-Cre⁻-RDTA;LysM-EGFP-ki; α-SMA-RFPcherry-Tg* (MC^ctrl) as controls (Fig. 2a). Applying confocal IVM to TNF-stimulated tissues, we noted that the frequency of neutrophil TEM

events was not significantly altered in MC^deficient animals as compared to control littermates (Fig. 2b, Supplementary Movie 6). In contrast, MC^deficient mice exhibited a substantial defect in neutrophil motility in the sub-EC space along pericyte processes. Specifically, neutrophils covered a longer distance (~82% increase), spent more time (~85% increase) and showed enhanced meandering behaviour (less straight migration) within the sub-EC space, as compared to neutrophil

**Fig. 1 | Perivascular MCs promote hotspots of neutrophil migration.**
**a–c** Cremaster muscles or ear skin of WT mice were stimulated with TNF (300 ng) or PBS for 4 h. Tissues were immunostained for neutrophils (MRP14), MCs (avidin) and pericytes/smooth muscle cells (α-SMA). **a** Mouse cremaster 4 h post TNF-stimulation, representative image of 5 independent experiments, scale bars 100 μm. **b, c** Correlation between the number of extravascular neutrophils and perivascular MCs (**b**, $n = 63$ venules, $p$ value ≤ 0.0001) or ear skin (**c**, $n = 41$ venules, $p$ value ≤ 0.001); data pooled from six mice. Line indicated linear regression and dashed lines 95% confidence band. **d–j** *LysM-EGFP-ki; α-SMA-RFPcherry-Tg* mice were stimulated with TNF (300 ng) for 2 h. AF532-labelled anti-CD31 and AF647-labelled anti-CD117 mAbs were injected i.s. to visualise EC junctions (dark blue) and MCs (cyan), respectively. **d** Illustration of the 4-colour confocal IVM methodology to simultaneously visualise the neutrophils, perivascular MCs, pericytes & ECs. Scale bars, 20 μm. **e** Scheme depicting neutrophil responses quantified. **f** Frequency of neutrophil TEM and TPM migrating <20 μm from a perivascular MC ($n = 5$ mice, $p$ value = 0.0191). **g** Images of TNF-stimulated post-capillary venules showing the last

time point of a neutrophil within the sub-EC space. Crawling path (coloured time-coded line) and directionality (white arrow) from site of neutrophil TEM to the site of venular wall exit distant to MC (left images) or adjacent to an MC (right images, see Movie S3); scale bars, 5 μm. **h** Neutrophil abluminal migration distance ($n = 17$ neutrophils distant to MC, $n = 14$ close to MC neutrophils, $p$ value = 0.0277), time ($n = 21$ neutrophils, $p$ value = 0.0482) and straightness ($n = 21/20$ neutrophils, $p$ value = 0.0014); data pooled from five mice. **i** Time lapse IVM images 2 h post TNF-stimulation (see Movie S5) illustrating a hotspot of neutrophil transpericyte migration (TPM) occurring in close apposition to MCs, interstitial view is shown, scale bars 7 μm. **j** Frequency of neutrophil TPM hotspot <10 μm from a perivascular MC or macrophage (Mac) or not related to perivascular immune cells ($n = 5$ mice, \*\*$p$ value = 0.006 #$p$ value = 0.0379). Mean ± SEM (each mouse represents one independent experiment). **b, c** Spearman's rank correlation test; **f, h** two-tailed Student's $t$ test; **j** one-way ANOVA followed by Tukey's post-hoc test. \*$p < 0.05$, \*\*$p < 0.01$ as compared to TEM, >10 or No Mac or MC or as indicated by #$p < 0.05$. Source data are provided as a Source Data file.

---

behaviours in stimulated venules of MC$^{ctrl}$ littermates (Fig. 2c, d, Supplementary Movie 7). Furthermore, the number of neutrophil TPM events, frequency of hotspots within the pericyte layer and number of tissue-infiltrated neutrophils were significantly reduced in MC$^{deficient}$ mice as compared to MC$^{ctrl}$ animals (Fig. 2e–g, Supplementary Movie 6).

Collectively, these results provide direct evidence for the ability of perivascular MCs to positively regulate neutrophil motility on pericytes and thus facilitate effective breaching of the venular wall.

## Interleukin-17A (IL-17A) is released from MCs upon inflammation and regulates neutrophil abluminal motility and exit of the pericyte layer

Having identified a role for perivascular MCs in neutrophil migration through the pericyte layer, we next sought to investigate the molecular basis of this response. We focussed our attention on IL-17A; a pro-inflammatory cytokine known to promote neutrophil recruitment in numerous inflammatory conditions[23] and expressed at transcript level by tracheal and dermal murine MCs, and by human MCs from RA synovium (www.immgen.org[24];). In initial works, we detected by flow cytometry the presence of IL-17A protein in single-cell-suspensions of MCs isolated from WT mouse cremaster muscles, ear skin, heart but not from the peritoneal cavity in basal conditions (Fig. 3a). IL-17A transcript was also detected in MCs isolated from cremaster muscles, indicating their capacity to produce this cytokine (Supplementary Fig. 4a). Interestingly, MC-IL-17A protein signal was reduced post TNF-stimulation of cremaster muscles (4 h; Fig. 3b, Supplementary Fig. 4b); and a similar observation was made in LPS-induced inflammation (Supplementary Fig. 4c); thus, indicating that MCs release of the cytokine upon inflammation. As the production of IL-17A has been associated with other leukocyte subtypes, we investigated potential alternative sources of the cytokine in our inflammatory model. Both IL-17A transcript and protein could be detected in tissue-resident macrophages (Fig. 3b and Supplementary Fig. 4a) but notably this expression was two-fold lower than the levels detected in MCs and was not altered upon TNF-stimulation (Fig. 3b). Furthermore, IL-17A was not detected in infiltrated neutrophils (Supplementary Fig. 4d), and since basophils and Th cells were not recruited to acutely inflamed tissues, these leukocytes were excluded as a source of IL-17A (Supplementary Fig. 4e, f).

To univocally ascertain MCs as a key cellular source of IL-17A, we compared the total level of this cytokine in control and inflamed cremaster muscles of MC$^{ctrl}$ and MC$^{deficient}$ mice (Fig. 3c). As observed in MCs of WT mice, levels of IL-17A in tissues of MC$^{ctl}$ animals was decreased following TNF stimulation as compared to PBS-treated animals (Fig. 3c). Importantly, MC$^{deficient}$ mice exhibited reduced levels of IL-17A in both control (PBS) (~64% reduction) and TNF-stimulated tissues, as compared to littermate controls (Fig. 3c). No

IL-17A was however detected in the plasma of mice subjected to TNF stimulation in both MC$^{ctrl}$ and MC$^{deficient}$ mice (Supplementary Fig. 4g), suggesting that MC-derived IL-17A likely exerts localised effects in the stroma.

To investigate the functional role of IL-17A in neutrophil trafficking in vivo, we compared the migration response of neutrophils in IL-17A deficient mice (IL-17A$^{KO}$) to that detected in control littermates (WT). Of note, the number of perivascular MCs in IL-17A$^{KO}$ mice was similar to that of WT mice in both control and inflamed tissues (Supplementary Fig. 5a). Functionally, IL-17A$^{KO}$ animals exhibited ~64% reduction of neutrophil extravasation at 4 h post-TNF-stimulation (Fig. 3d, e) and we noted a loss of correlation between perivascular MC numbers and tissue-infiltrated neutrophils ($r^2 = 0.003$, Supplementary Fig. 5b) as compared to WT animals (Fig. 1b). We next sought to investigate if IL-17A controlled neutrophil-pericyte interactions in vivo, a hypothesis supported by in vitro findings showing that IL-17A can activate cultured human pericytes but not ECs (i.e., HUVECs and human dermal microvascular ECs)[8]. Initially, to ascertain whether pericytes can respond to IL-17A in vivo, the expression profile of its receptor, IL-17RA, was analysed by flow cytometry in single-cell suspensions of pericytes, as compared to ECs and macrophages, isolated from the lungs, ear skin and cremaster muscles of *α-SMA-RFPcherry-Tg* animals. We found that irrespective of their tissue source, pericytes expressed high surface levels of IL-17RA as compared to ECs and macrophages (Fig. 3f and Supplementary Fig. 6), suggesting that pericytes could be a significant and preferential target for IL-17A.

Finally, using our 4-colour IVM approach, we assessed the effect of the neutralising anti-IL-17A mAb on neutrophil diapedesis in real time. Whilst the number of neutrophil TEM events were not impacted by local treatment with the anti-IL-17A blocking mAb (Fig. 3g Supplementary Movies 8 and 9), the antibody induced a significant disruption of neutrophil motility in the sub-EC space, as exemplified by an increase in leukocyte abluminal crawling length and duration (~50% and ~107% increase, respectively) and reduced straightness (~67% decrease), as compared to responses detected in tissues treated with a control Ab (Fig. 3h). In addition, both the number of neutrophils exiting the pericyte layer and the number of TPM hotspots were significantly reduced in anti-IL-17A mAb treated mice, resulting in ~69% and ~75% reductions, respectively (Fig. 3i, j). Together, these responses were associated with a significant suppression of total neutrophil extravasation into inflamed tissues (Fig. 3k). Furthermore, using a transwell migration assay in vitro, we observed that TNF-stimulated MCs promoted the migration of neutrophils through a layer of pericytes, a response partially inhibited in the presence of the neutralising anti-IL-17A antibody (Supplementary Fig 7).

Collectively, our data identify perivascular MCs as a significant source of IL-17A and indicate its release following stimulation. Furthermore, this cytokine drives the direction and intensity of neutrophil

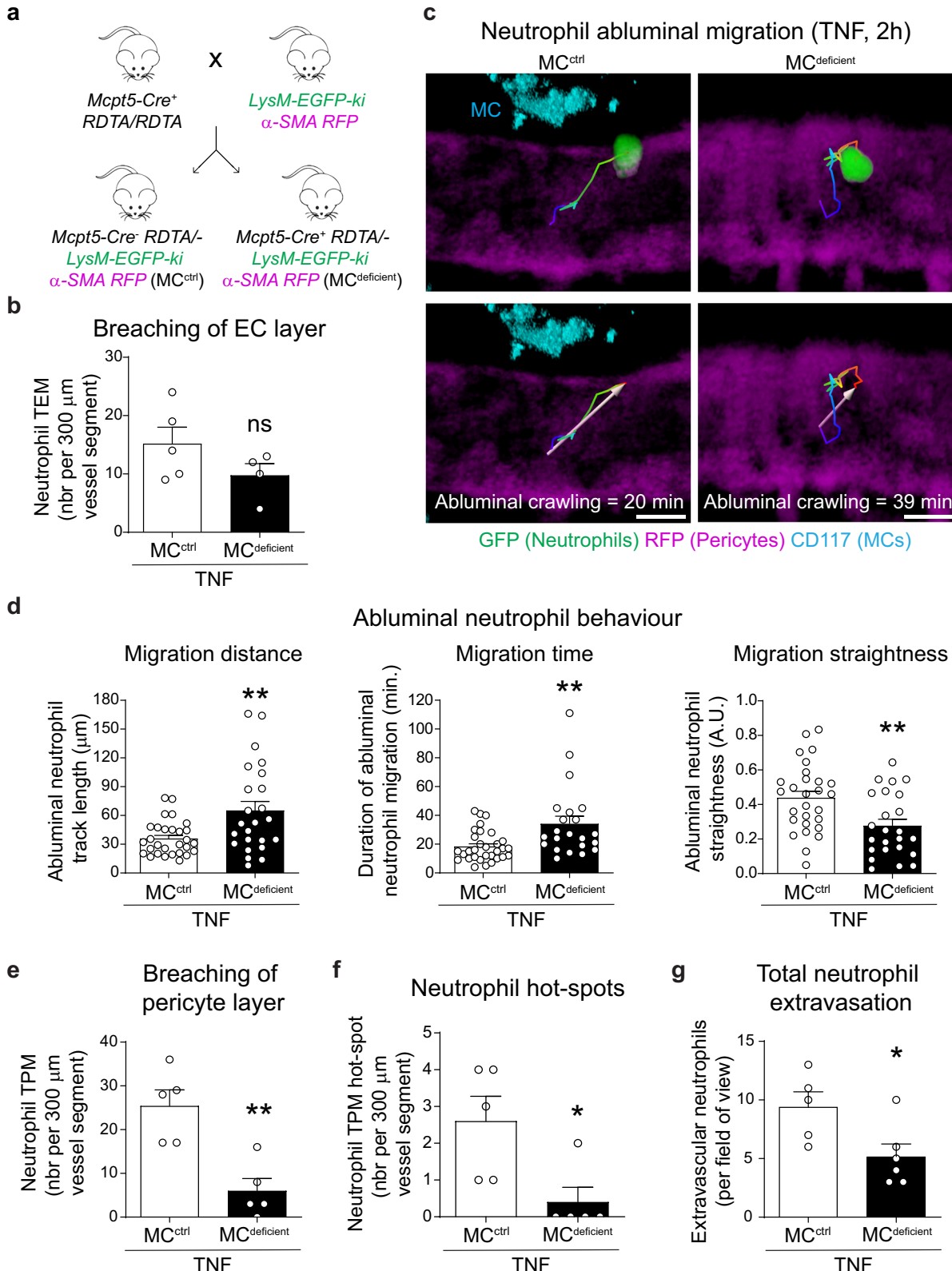

a

Mcpt5-Cre⁺
RDTA/RDTA **X** *LysM-EGFP-ki*
*α-SMA RFP*

*Mcpt5-Cre⁻ RDTA/-*
*LysM-EGFP-ki*
*α-SMA RFP* (MCᶜᵗʳˡ) *Mcpt5-Cre⁺ RDTA/-*
*LysM-EGFP-ki*
*α-SMA RFP* (MCᵈᵉᶠⁱᶜⁱᵉⁿᵗ)

b

**Breaching of EC layer**

Neutrophil TEM (nbr per 300 μm vessel segment)

ns

MCᶜᵗʳˡ MCᵈᵉᶠⁱᶜⁱᵉⁿᵗ
TNF

c

**Neutrophil abluminal migration (TNF, 2h)**

MCᶜᵗʳˡ MCᵈᵉᶠⁱᶜⁱᵉⁿᵗ

MC

Abluminal crawling = 20 min Abluminal crawling = 39 min

GFP (Neutrophils) RFP (Pericytes) CD117 (MCs)

d

**Abluminal neutrophil behaviour**

**Migration distance** **Migration time** **Migration straightness**

Abluminal neutrophil track length (μm) — ** — MCᶜᵗʳˡ MCᵈᵉᶠⁱᶜⁱᵉⁿᵗ TNF

Duration of abluminal neutrophil migration (min.) — ** — MCᶜᵗʳˡ MCᵈᵉᶠⁱᶜⁱᵉⁿᵗ TNF

Abluminal neutrophil straightness (A.U.) — ** — MCᶜᵗʳˡ MCᵈᵉᶠⁱᶜⁱᵉⁿᵗ TNF

e

**Breaching of pericyte layer**

Neutrophil TPM (nbr per 300 μm vessel segment) — ** — MCᶜᵗʳˡ MCᵈᵉᶠⁱᶜⁱᵉⁿᵗ TNF

f

**Neutrophil hot-spots**

Neutrophil TPM hot-spot (nbr per 300 μm vessel segment) — * — MCᶜᵗʳˡ MCᵈᵉᶠⁱᶜⁱᵉⁿᵗ TNF

g

**Total neutrophil extravasation**

Extravascular neutrophils (per field of view) — * — MCᶜᵗʳˡ MCᵈᵉᶠⁱᶜⁱᵉⁿᵗ TNF

trafficking in the sub-EC space and final exit of neutrophils through venular walls.

**Perivascular MCs and IL-17A promote a graded expression of ICAM-1 and CXCL1 on pericytes in vivo**

To explore the molecular basis of IL-17A-dependent neutrophil motility on pericytes, we focussed our attention on key drivers of this response, namely pericyte-associated ICAM-1 and CXCL1[9,10]. Hypothesising that IL-17A can regulate the expression of these effector molecules, we tested the effect of exogenous IL-17A, as compared to TNF, on ICAM-1 and CXCL1 expression levels on cultured primary pericytes (Supplementary Fig. 8). Here, stimulation of pericytes with IL-17A alone (6 h) led to a modest but significant increase in expression of ICAM-1 in a dose-dependent manner (Supplementary Fig. 8a, b) with TNF

**Fig. 2 | Abluminal migration of neutrophils is impaired in MC$^{deficient}$ animals.**
**a** Generation of MC-deficient mice exhibiting fluorescent neutrophils and pericytes
(*LysM-EGFP-ki; α-SMA-RFPcherry-Tg mice*). **b**–**g** MC$^{deficient}$ or MC$^{ctrl}$ animals were
subjected to TNF-induced (300 ng) inflammation for 2 h and analysed by confocal
IVM. AF532-labelled anti-CD31 and AF647-labelled anti-CD117 mAbs were injected
i.s. to visualise EC junctions and MCs (cyan), respectively. **b** Number of neutrophil
TEM events (*n* = 5 MC$^{ctrl}$ and *n* = 4 MC$^{deficient}$). **c** Neutrophil abluminal crawling paths
(pseudo-coloured lines) and directionality (white arrow) from the site of TEM to the
site of TPM in MC$^{ctrl}$ (left images) and MC$^{deficient}$ (right images, see Movie S7) mice;
pseudo-coloured line indicates time scale (from blue: TEM to red: TPM), scale bars,

5 μm. **d** Neutrophil abluminal migration distance (*n* = 29 MC$^{ctrl}$ and *n* = 24 MC$^{deficient}$
neutrophils, *p* value = 0.0026), time (*n* = 31 MC$^{ctrl}$ and *n* = 22 MC$^{deficient}$ neutrophils,
*p* value = 0.025) and straightness (*n* = 28 MC$^{ctrl}$ and *n* = 24 MC$^{deficient}$ neutrophils,
*p* value = 0.0037); data pooled from five mice. **e** Numbers of neutrophil TPM events
(*n* = 5 mice per group, *p* value = 0.0031), **f** neutrophil TPM hotspot (*n* = 5 mice per
group, *p* value = 0.0234) and **g** extravascular neutrophils, more than 20 μm from
venular wall (*n* = 5 MC$^{ctrl}$ and *n* = 6 MC$^{deficient}$ mice *p* value = 0.0316). Mean ± SEM
(each mouse represents one independent experiment). **b**, **d**–**g** two-tailed Student's
*t* test. *\**p* < 0.05, \*\**p* < 0.01 as compared to PBS or MC$^{ctrl}$ (ns = not significant). Source
data are provided as a Source Data file.

exerting a stronger response as compared to IL-17A. Both IL-17A and
TNF also induced release of CXCL1 by cultured pericytes in the
supernatant (Supplementary Fig. 8c).

In vivo, in line with our previous reports[9,10], total pericyte ICAM-1
and CXCL1 were increased in TNF-stimulated cremaster muscles
(Fig. 4a, b). However, these responses were still observed under con-
ditions of MC deficiency, IL-17A genetic deletion or pharmacological
blockade (Fig. 4c–f and Supplementary Fig. 9).

Considering the heterogenous distribution of MCs within cre-
master tissues (Fig. 1 and Supplementary Fig. 1a), we hypothesised that
MC-derived IL-17A may impact locally the distribution of pericyte-
associated ICAM-1 and CXCL1 in vivo whilst not changing their overall
expression level following TNF-stimulation. Hence, we used high-
resolution deconvoluted confocal imaging to investigate in more
detail the patterned expression of pericyte-associated ICAM-1 and
CXCL1. Intriguingly, we observed that whilst evenly distributed in TNF-
stimulated MC-free venular regions, stimulated MC-enriched venular
segments exhibited an enhanced pericyte ICAM-1 expression signal in
close proximity (<10 μm) to MCs (Fig. 5a). Furthermore, whilst TNF-
induced upregulation of pericyte ICAM-1 was observed in MC-free
venular segments, this response was higher in MC-enriched venules
(exhibiting more than three perivascular MCs) (Fig. 5b). Similar
observations were made for pericyte CXCL1 expression (Fig. 5c).

Together, these data suggest that perivascular MCs may regulate
local expression of ICAM-1 and CXCL1. To test this hypothesis, we
quantified in more detail the spatial distribution of these molecules
within the pericyte sheath of unstimulated and stimulated WT venules
every 5 μm away from the nearest perivascular MC. With this approach,
whilst the signal was similar across the whole pericyte surface in
unstimulated tissues, following TNF stimulation, pericyte ICAM-1 MFI
showed an increased expression towards MCs (-28% increased), as
compared to the more distal region (Fig. 5d). This graded expression
was absent in IL-17A$^{KO}$ animals (Fig. 5e) and in WT animals treated with
the anti-IL-17A blocking antibody (Fig. 5f). Similar observations were
made for pericyte CXCL1 expression (Fig. 5g–i). Interestingly, a graded
expression of ICAM-1 and CXCL1 on pericytes near MCs was also
observed following LPS-induced inflammation of cremaster tissues in
WT but not IL-17A$^{KO}$ mice. Furthermore, total neutrophil extravasation
was significantly reduced (-60% reduction) in IL17-A$^{KO}$ as compared to
WT animals upon LPS stimulation (Supplementary Fig. 10).

Together, these data demonstrate that localised expressions of
ICAM-1 and CXCL1 on the pericyte layer is regulated by MCs and IL-17A
upon acute inflammation.

## MC-derived IL-17A drives ICAM-1 and CXCL1 graded expression on pericytes

In a final series of experiments, we sought to directly investigate the
role of MC-derived IL-17A in regulating pericyte phenotype and neu-
trophil extravasation. For this purpose, using previously detailed
methods[25] we reconstituted MC$^{deficient}$ mice with bone marrow derived
MCs (BMMCs) from either WT or IL-17A$^{KO}$ donor animals. This
approach generated mice with WT MCs (MC$^{WT}$) and mice with IL-17A
deficient MCs (MC$^{IL-17AKO}$), respectively (Fig. 6a). Of note, both MC$^{WT}$ or
MC$^{IL-17AKO}$ animals showed similar number and perivascular localisation

of donor MCs in the cremaster muscles (Fig. 6b, c), indicating an effi-
cient reconstitution of tissue-connective MCs. Following TNF-stimu-
lation, we did not observe a significant change in the total levels of
pericyte-associated ICAM-1 and CXCL1 in MC$^{IL-17AKO}$ mice, as compared
to MC$^{WT}$ animals (Fig. 6d, e). Importantly, the graded expression of
pericyte ICAM-1 and CXCL1 detected in MC$^{WT}$ mice was absent in MC$^{IL-17AKO}$ animals (Fig. 6f–h). In line with these findings, total neutrophil
extravasation was significantly reduced in MC$^{IL-17AKO}$ mice (-36% inhi-
bition, as compared to MC$^{WT}$ mice; Fig. 6i) and we observed no cor-
relation between perivascular MCs and tissue-infiltrated neutrophil
numbers in MC$^{IL-17AKO}$ mice (*r*$^2$ = 0.0005, *P* value = 0.86) as compared to
MC$^{WT}$ animals (*r*$^2$ = 0.17, *P* value= 0.006) (Fig. 6j, k).

In summary, these results demonstrate that MC-derived IL-17A is
instrumental in establishing vascular regions characterised by
enhanced expressions of ICAM-1 and CXCL1 on pericytes, an effect that
promotes efficient diapedesis of circulating neutrophils in vivo.

## Discussion

Neutrophil diapedesis is controlled by unique molecular cues that are
spatially and temporally expressed by distinct cellular components of
venular walls, namely ECs and pericytes[2,26]. Despite extensive studies
deciphering the mechanisms of neutrophil interactions with ECs,
details of the cellular and molecular signals that control neutrophil
trafficking from the sub-endothelial space into the interstitium remain
unclear. Here, we describe a crucial role for perivascular MCs in con-
trolling the motility of neutrophils within the pericyte sheath, facil-
itating their exit from the vessel wall in close apposition to MC-
enriched regions. This response is achieved through altered expres-
sion and patterning of key regulatory molecules on pericytes as
mediated by MC-derived IL-17A (Fig. 7). The discovery of this axis adds
a new component to the cascade of molecular and cellular mechan-
isms that mediate neutrophil recruitment and identifies MCs and
pericytes as potential targets for therapeutic interventions aimed at
modulating inflammation.

Sentinel leukocytes residing in the interstitium are instrumental in
the initiation and propagation of inflammatory responses following
pathogen invasion or tissue injury[27]. Specifically, tissue-resident mac-
rophages and perivascular MCs control the rapid recruitment of neu-
trophils from the circulation through the release of pro-inflammatory
cytokines that activate the endothelium and neutrophil
chemoattractants[16]. Here, using a TNF-induced acute inflammatory
model, intense neutrophil diapedesis was directly associated with
perivascular MCs, as demonstrated by a reduction of neutrophil
recruitment in MC-deficient animals. In contrast, no association
between sites of neutrophil extravasation and perivascular macro-
phages could be detected, despite their higher total numbers and
more even distribution pattern in the perivascular compartments.
These observations are supported by a previous study demonstrating
that in LPS-driven peritonitis, MCs but not macrophages, secrete
CXCL1 to promote neutrophil exit from vessel walls[16]. Furthermore, the
influence of MC-derived CXCL1 on neutrophil diapedesis was drama-
tically demonstrated in IL1β-stimulated aged tissues where excessive
production of CXCL1 by perivascular MCs led to dysregulated neu-
trophil TEM[28]. Here, we found that CXCL1 signal detected in

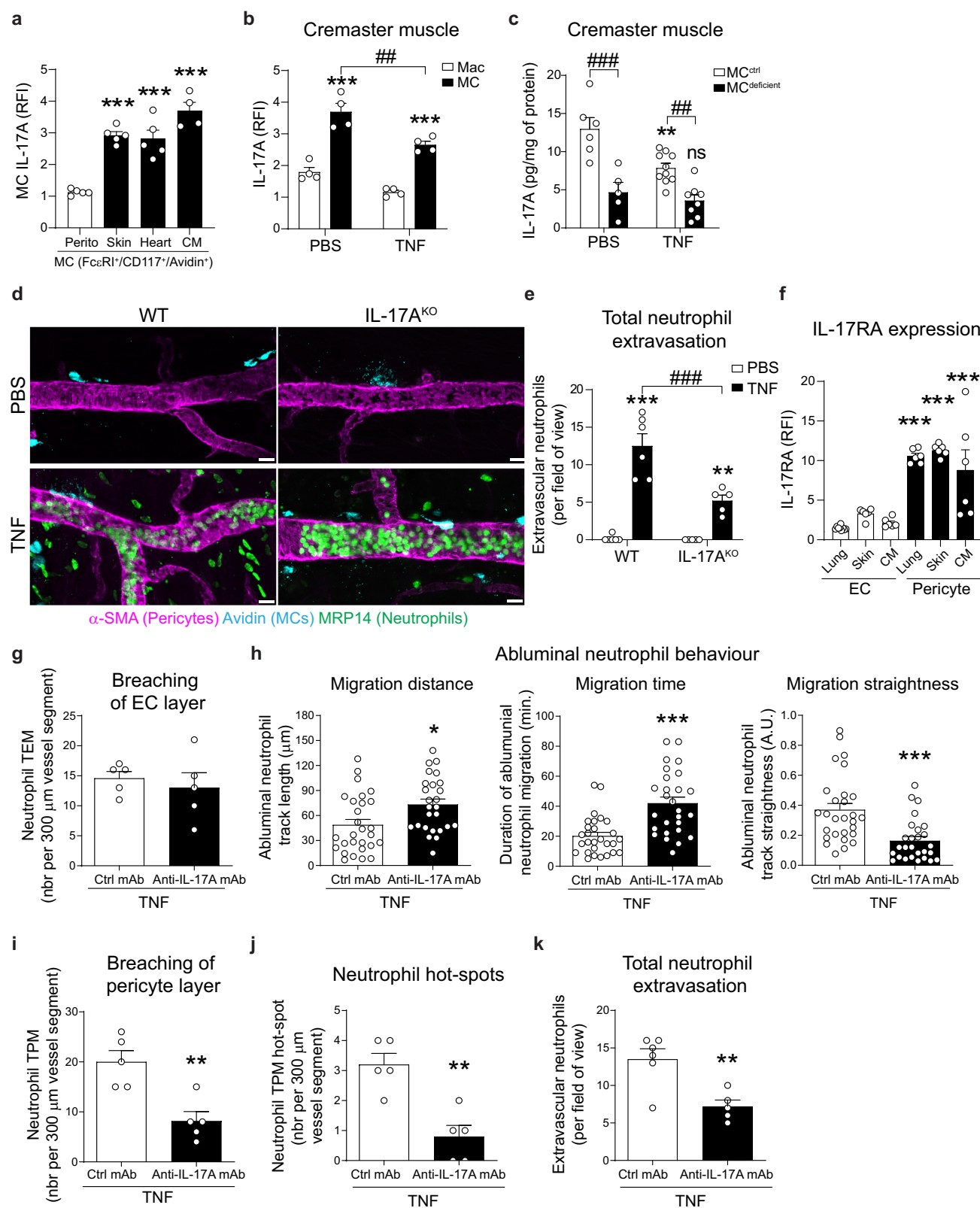

α-SMA (Pericytes) Avidin (MCs) MRP14 (Neutrophils)

perivascular MCs was significantly increased upon TNF-induced inflammation of the cremaster muscles (Supplementary Fig. 11a). However, the total levels of CXCL1 in tissues were similar between WT and MC$^{deficient}$ animals (Supplementary Fig. 11b), indicating the role of other stroma cells for CXCL1 production in this inflammatory model. Furthermore, in skin infection models of *Staphylococcus aureus*, perivascular macrophages controlled neutrophil recruitment and

regulated their interstitial motility through release of inflammatory chemokines[29]. These findings suggest the existence of diverse, yet specific mechanisms of neutrophil recruitment as controlled by distinct sentinel leukocytes in a tissue-dependent manner and/or as driven by different inflammatory scenarios.

Whilst a minor fraction of the immune microenvironment, MCs are located in all tissues of the body, exhibiting a heterogenous

**Fig. 3 | Mast cells release IL-17A upon TNF-stimulation and IL-17A promotes neutrophil abluminal migration and final exit. a** Cremaster muscles, ear skin, heart and peritoneal lavage (perito) from WT mice was collected and analyse for IL-17A expression in MCs. Quantification of IL-17A expression by RFI in MCs ($n = 5$ perito, $n = 5$ skin, $n = 5$ heart and $n = 4$ CM, all $p$ value < 0.0001). **b** Quantification of IL-17A expression by RFI in MC and macrophages (CD45$^+$, CD115$^+$) in PBS and TNF treated (4 h) WT mice ($n = 4$ mice, ***$p$ value < 0.0001, ##$p$ value = 0.002). **c** Tissue level of IL-17A analysed by ELISA in PBS and TNF treated (4 h) MC$^{ctrl}$ or MC$^{deficient}$ ($n = 6$ PBS MC$^{ctrl}$ mice, $n = 5$ PBS MC$^{deficient}$ mice, $n = 10$ TNF MC$^{ctrl}$ mice and $n = 8$ TNF MC$^{deficient}$ mice, **$p$ value = 0.0015, ###$p$ value < 0.0001, ##$p$ valu e =0.0018). **d** Images of the cremasteric microcirculation 4 h post TNF or PBS treatment in WT and IL-17A$^{KO}$ mice, scale bars 10 μm. **e** Number of extravascular neutrophils ($n = 6$ PBS WT mice, $n = 5$ PBS IL-17A$^{KO}$ mice, $n = 6$ TNF WT mice and $n = 5$ IL-17A$^{KO}$ mice, ***$p$ value < 0.0001, **$p$ value = 0.002, ###$p$ value < 0.0001). **f** Quantification of IL-17RA expression by RFI in indicated cell populations and organs, full gating strategy is

shown in Supplementary Fig. 6 ($n = 6$ mice, all $p$ value < 0.0001). **g–k** *LysM-EGFP-ki; α-SMA-RFPcherry-Tg* mice were subjected to TNF (300 ng) induced inflammation for 2 h. Blocking anti-IL-17A mAb (50 μg) or control mAb, was injected i.s. together with TNF. **g** Number of neutrophil TEM events ($n = 5$ mice). **h** Neutrophil abluminal migration distance ($n = 28$ Ctrl Mab $n = 26$ anti-IL17A mab neutrophils, $p$ value = 0.010), time ($n = 28$ neutrophils, $p$ value < 0.0001) and straightness ($n = 28$ neutrophils, $p$ value = 0.0002); data pooled from five mice. **i** Number of neutrophil TPM events ($p$ value = 0.0037), **j** neutrophil TPM hotspots ($p$ value 0.0019) and **k** number of extravascular neutrophils ($p$ value = 0.0051) ($n = 5$ mice). Mean ± SEM (each mouse represents one independent experiment). **a, f** one-way ANOVA followed by Tukey's post-hoc test; **b, c, e** two-way ANOVA followed by Sidak's post-hoc test; **g-k** two-tailed Student's $t$ test. *$p$ < 0.05, **$p$ < 0.01, ***$p$ < 0.001 as compared to perito, Mac, MC$^{ctrl}$, PBS, ECs or ctrl mAb or as indicated by ##$p$ < 0.01, ###$p$ < 0.001 (ns = not significant). Source data are provided as a Source Data file.

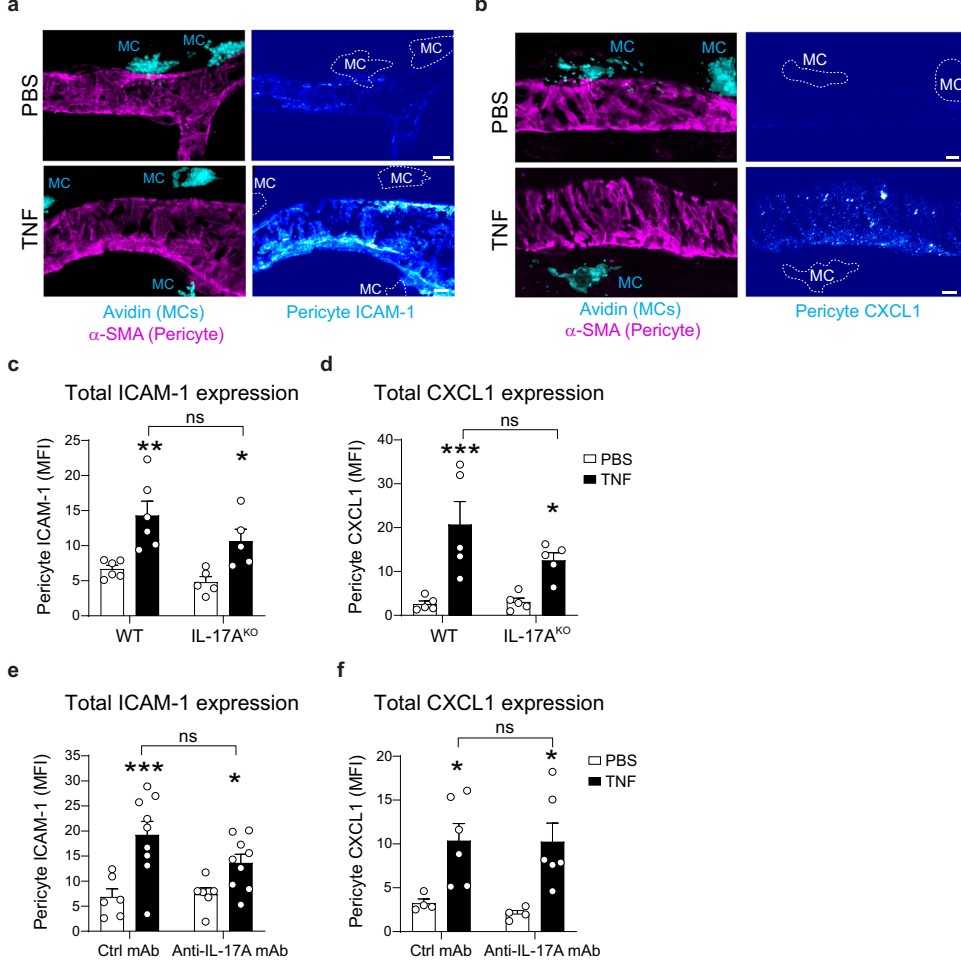

**Fig. 4 | Pericyte ICAM-1 and CXCL1 expression is increased in vessels with MCs but total expression is not impacted by IL-17A. a–d** Cremaster muscles of WT or IL-17A$^{KO}$ mice were stimulated with TNF (300 ng) or PBS for 4 h. Cremaster muscles were collected and immunostained for MCs (avidin), pericytes (α-SMA) and ICAM-1 or CXCL1. **a–b** Images of venules showing **a** ICAM-1 or **b** CXCL1 expression (pseudocolour intensity gradient) on pericytes (purple) and MCs (cyan), dashed lines the limit of MC, scale bars, 5 μm. **c, d** Quantification of **c** ICAM-1 MFI ($n = 6$ WT, $n = 5$ IL-17A$^{KO}$ mice) and **d** CXCL1 MFI ($n = 5$ mice) on pericytes. **e, f** Cremaster muscles of

WT mice were stimulated with TNF (300 ng) or PBS in combination with a blocking anti-IL-17A or control mAbs (50 μg) for 4 h. **e, f** Quantification of **e** ICAM-1 MFI ($n = 6$ PBS Ctrl mAb, $n = 9$ TNF Ctrl mAb, $n = 6$ PBS Anti-IL-17A mAb, $n = 9$ TNF anti-IL-17A mAb mice) and **f** CXCL1 MFI ($n = 4$ PBS Ctrl mAb, $n = 6$ TNF Ctrl mAb, $n = 4$ PBS Anti-IL-17A mAb, $n = 6$ TNF anti-IL-17A mAb mice). Mean ± SEM (each mouse represents one independent experiment). **c–f** two-way ANOVA followed by Sidak's post-hoc test. *$p$ < 0.05, **$p$ < 0.01, ***$p$ < 0.001 as compared to WT or ctrl mAb (ns = not significant). Source data are provided as a Source Data file.

distribution and diverse phenotype in an organ-specific manner. Here, in the cremaster muscle, they represent ~25% of total tissue-resident leukocytes. Interestingly, most MCs show a perivascular localisation -in particular post-capillary venules and arterioles- that supports their well-known role in modulating vascular tone and permeability,

responses extensively studied in the context of allergic and hypersensitivity reactions[30–32]. In the ear skin MC density is equivalent to the number of dermal dendritic cells and more evenly distributed[33]. The tissue-dependent distribution of MCs is suggested to depend on the migration of MC progenitors from the yolk sac during early

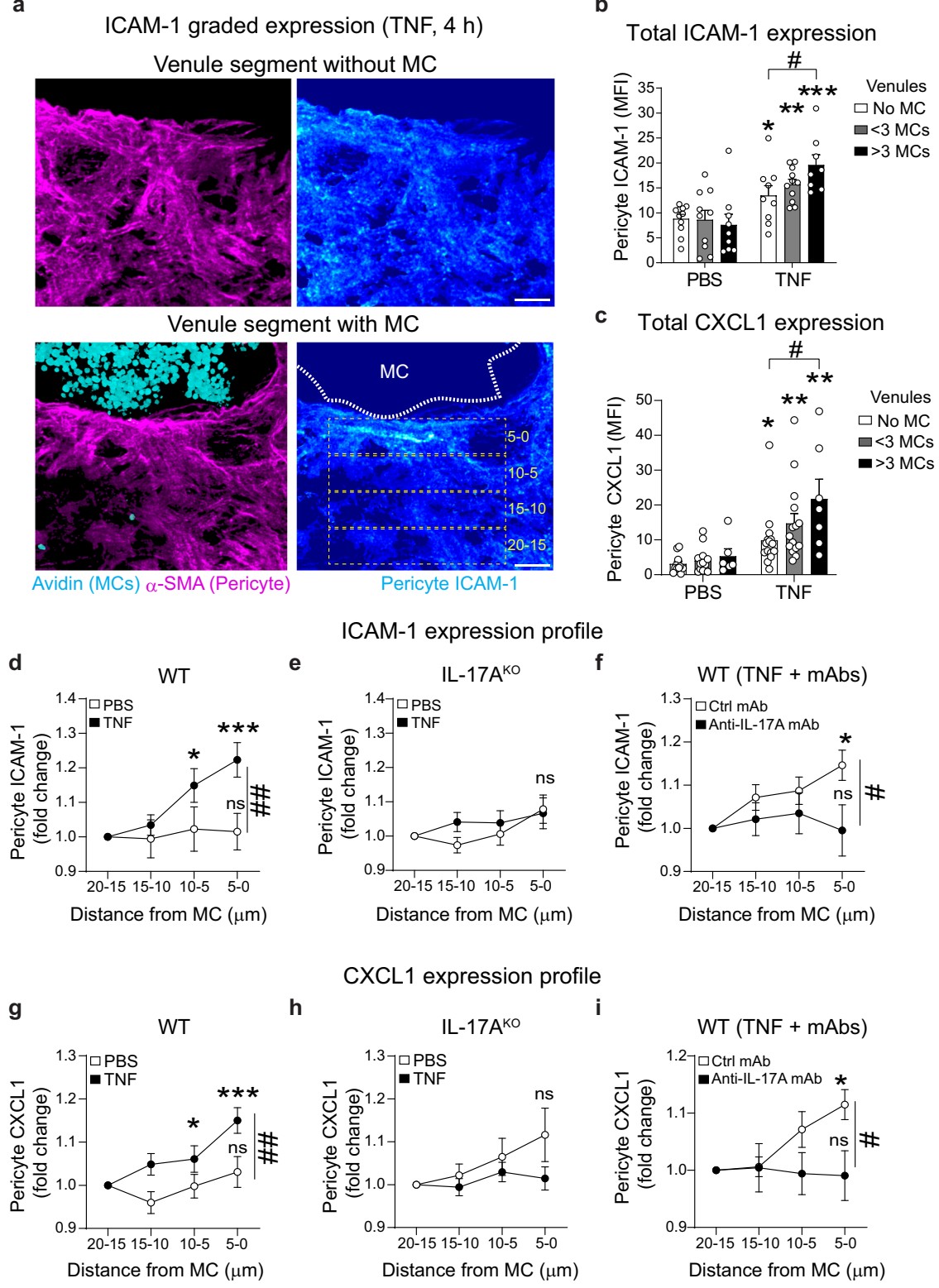

embryogenesis and the moderate proliferative capacity of mature MCs in vivo[34]. Mechanistically, the migration of these progenitors and definite localisation of MCs in peripheral tissues are still not fully understood. However, a recent study by Bambach et al. demonstrated that in the mouse dermis, c-KIT ligand/Stem cell factor (SCF, key mediator of maturation, development and migration of MCs) were produced by stroma cells of periarteriolar wall (i.e., EC and vascular smooth muscles cells) and that β1-integrins controlled the migration and positioning of MCs along arteriolar walls[35]. Whilst connective MC

heterogenous distribution along PCVs is still unclear, publicly available RNAseq data indicate that pericytes do also express SCF[36]. Interestingly, here we observed that during acute inflammation, the heterogenous distribution of MCs along post-capillary venules of murine cremaster muscles was associated with the formation of "hotspots" of neutrophil migration through pericytes. The phenomena of hotspots have been studied in detail in the context of neutrophil breaching of the endothelium[37–40], but less is known about the molecular basis of its occurrence across the pericyte layer. Specifically, hotspots of

**Fig. 5 | MCs promote graded expression of ICAM-1 and CXCL1 on pericytes.**
**a**–**d** TNF-stimulated or PBS-treated WT cremaster muscles were immunostained for MCs (avidin), pericytes (α-SMA) and ICAM-1 or CXCL1. **a** High magnification confocal image of a TNF-stimulated post-capillary venule (halved) showing enrichment of ICAM-1 (pseudocolour intensity) associated with pericytes (purple) in venule segment with or without MCs (cyan). White dashed line shows the limit of MC, scale bars, 5 μm. **b, c** Quantification of pericyte **b** ICAM-1 ($n = 11$ PBS no MC, $n = 10$ PBS < 3 MCs, $n = 9 > 3$ MCs, $n = 9$ TNF no MC, $n = 12$ TNF < 3 MCs, $n = 8$ TNF > 3 MCs, *$p$ value = 0.0451, **$p$ value = 0.0047, ***$p$ value<0.0001, #$p$ value = 0.0486) or **c** CXCL1 ($n = 12$ PBS no MC, $n = 13$ PBS < 3 MCs, $n = 7 > 3$ MCs, $n = 15$ TNF no MC, $n = 15$ TNF < 3 MCs, $n = 7$ TNF > 3 MCs, *$p$ value = 0.0411, **$p$ value = 0.0028, **$p$ value < 0.0024, #$p$ value = 0.0087) MFI in venules without, less than or more than 3 MCs. **d**–**f** Quantification of pericyte ICAM-1 MFI in 5 μm-wide consecutive regions (as exemplified in **a**) from a perivascular MC in **d** PBS and TNF treated WT mice ($n = 18$ PBS, 31 TNF perivascular MC regions, data pooled from five mice, *$p$ value = 0.0366, ***$p$ value = 0.003, ##$p$ value = 0.0044), **e** PBS and TNF treated IL-17A$^{KO}$ mice ($n = 36$ perivascular MC regions, data pooled from five mice) or **f** in TNF treated mice in combination with ctrl mAb or anti-IL-17A mAb ($n = 27$ Ctrl mAb, 28 anti-IL-17A mAb perivascular MC regions, *$p$ value = 0.0265, #$p$ value = 0.0133); data pooled from four mice. **g**–**i** Quantification of pericyte CXCL1 MFI in 5 μm-wide consecutive regions from a perivascular MC in **g** PBS and TNF-treated WT mice ($n = 31$ PBS, 30 TNF perivascular MC regions, data pooled from five mice, *$p$ value = 0.0322, ***$p$ value = 0.003, ##$p$ value = 0.0041), **h** PBS and TNF treated IL-17A$^{KO}$ mice ($n = 41$ perivascular MC regions, data pooled from five mice) or **i** in TNF treated mice in combination with ctrl mAb or anti-IL-17A mAb ($n = 21$ Ctrl mAb, 27 anti-IL-17A mAb perivascular MC regions *$p$ value = 0.0248, #$p$ value = 0.0448); data pooled from four mice. ICAM-1 and CXCL1 MFI were normalised to the most distal region from the MC (i.e., 15–20 μm). Mean ± SEM (each mouse represents one independent experiment). **b**–**i** two-way ANOVA followed by Sidak's post-hoc test. *$p < 0.05$, **$p < 0.01$, ***$p < 0.001$ as compared to PBS or 20–15 region or as indicated by #$p < 0.05$, ##$p < 0.01$ (ns = not significant). Source data are provided as a Source Data file.

neutrophil migration through the endothelium have been associated with regulated expression of β2-integrins on neutrophils[39], compromised endothelial cell autophagy[40], and formation of endothelial junctional membrane protrusions enriched in adhesion molecules (i.e., ACKR1, ICAM-1,CD31)[37]. Here, building on our previous works of neutrophil hotspots through the pericyte layer[10], we report on the establishment of this response near perivascular MCs. Seemingly an enigma, hotspots are a notable feature of leukocyte diapedesis that may minimise the disruption of venular walls and promote neutrophil recruitment close to specific sites of infection/inflammation within tissues[39,41]. As such, the molecular basis of this event, especially at the level of pericytes, warrants further mechanistic analysis as addressed here.

Several MC-derived mediators, including cytokines (e.g., TNF, IL-1β, IL-6, GM-CSF)[17,18,42–44], proteases (e.g., MCP-6[45]) and chemoattractants (e.g., CXCL1, CCL2) are implicated in neutrophil recruitment in models of sterile and non-sterile inflammation. Interestingly, a limited number of studies reported that MCs can produce IL-17A[24,46]. Classically associated with γδ T lymphocytes, ILCs and Th17 cells[47], IL-17A is a key cytokine regulating physiological and protective inflammatory responses against infections. IL-17A is also intimately implicated in the development of TNF-driven autoimmune disorders such as psoriasis, spondyloarthritis and Crohn's disease[48]. Furthermore, MCs are the predominant cellular source of IL-17A in the synovium of rheumatoid arthritis patients[24]. As such, anti-IL17A-targeting drugs are now proposed to patients with severe psoriasis and psoriatic arthritis[49,50]. Here we noted that whilst MCs from cremaster muscles constitutively expressed IL-17A at steady-state, they released this cytokine upon TNF-induced inflammation. Furthermore, MC$^{deficient}$ mice exhibited significant reduction in total IL-17A levels in inflamed tissues. Of note, whilst no T cells or basophils could be detected in acutely inflamed tissues, we did not detect IL-17A in infiltrated neutrophils; as supported by the literature[51]. Although the precise subcellular localisation of IL-17A in MCs remains elusive, this cytokine may be secreted post TNF-stimulation through piecemeal exocytosis[52]. Whilst studies have shown that MC-derived factors such as TNF can diffuse into the systemic circulation in hapten-induced skin inflammation[13] and affect neutrophil recruitment, here we did not detect the presence of IL-17A in the serum of WT or MC$^{deficient}$ mice under basal or TNF-stimulated conditions (Supplementary Fig. 4g). This would suggest that MC-derived IL-17A may act locally.

When released, IL-17A acts through its principal receptor, IL-17RA, to promote transcription of pro-inflammatory genes regulated by NFκB and Map-kinase pathways[53]. Here, we demonstrated that pericytes expressed high levels of the IL-17A receptor as compared to ECs; whilst this receptor was absent from the surface of tissue-resident macrophages (with or without inflammation). Thus, these data identified pericytes as a key cellular target of MC-derived IL-17A.

In the literature, IL-17A was reported to synergise with TNF in enhancing gene transcripts for CXCL8, CXCL5 and CCL20 on human pericytes but not on ECs in vitro[8]. We noted that IL-17A stimulated the upregulation of ICAM-1 and release of CXCL1 by cultured pericytes isolated from murine cremaster muscles. Surprisingly, pharmacological blockade or genetic deficiency in IL-17A did not alter the total expression levels of these effector molecules on TNF-stimulated pericytes in vivo, suggesting no synergistic effect between the two cytokines in our inflammatory model. Nevertheless, inhibition of IL-17A signalling led to altered motility of neutrophils in the sub-EC space, reduced occurrence of transpericyte hotspots and inhibition of total tissue-infiltration of neutrophils. Whilst we previously demonstrated that TNF promoted CXCL1 and ICAM-1 upregulation by pericytes both in vitro and in vivo[10], control of neutrophil motility in the pericyte layer and exit from the venular wall required further clarifications. In this context, our group and others have shown the critical importance of adhesion molecules and chemokines in modulation of neutrophil sub-EC motility[10,39]. Specifically, we reported the enrichment of ICAM-1 and CXCL1 around gaps in the pericyte sheath that were preferentially used by neutrophils to breach the pericyte layer[10]. These results suggested that regulated expression of key molecules could play a role in directing neutrophils within the sub-EC space towards exit sites in the venular wall. Extending these observations, the findings of the present study revealed that in TNF-stimulated tissues, pericytes show a graded expression of ICAM-1 and CXCL1 in venular segments enriched with perivascular MCs. Most importantly, we noted that the enhanced levels of ICAM-1 and CXCL1 on pericytes close to MCs was IL-17A-dependent. Although the mechanism leading to this finely tuned modulation of pericyte phenotype is not fully understood, IL-17A can promote cytoskeletal rearrangement of smooth muscles cells via PKC and RhoA/ROCK2 activation[54,55]. As these signalling molecules are involved in cell surface reorganisation of membrane- and cytoskeleton-bound molecules, such mechanism could also contribute to spatial reorganisation of ICAM-1 and CXCL1 on pericytes. The latter could lead to the establishment of a molecular path that directs neutrophils within the sub-EC space towards the nearest perivascular MCs. Similar modulation of endothelial ICAM-1 and ACKR1 (a receptor known to present CXCL1 to neutrophils) distribution has been associated with localised neutrophil TEM[41].

Taken together, the present findings shed light on the role of MCs and IL-17A in the final stages of neutrophil diapedesis, sub-EC motility and breaching of the pericyte layer. The physiological role of the localised site of neutrophil extravasation (hotspots) remains enigmatic but is reminiscent of other forms of coordinated migration behaviours, such as neutrophil interstitial swarming towards sites of injury[56]. The directed sub-EC migration of neutrophils toward MCs offers a mechanism for efficient breaching of venular walls and a

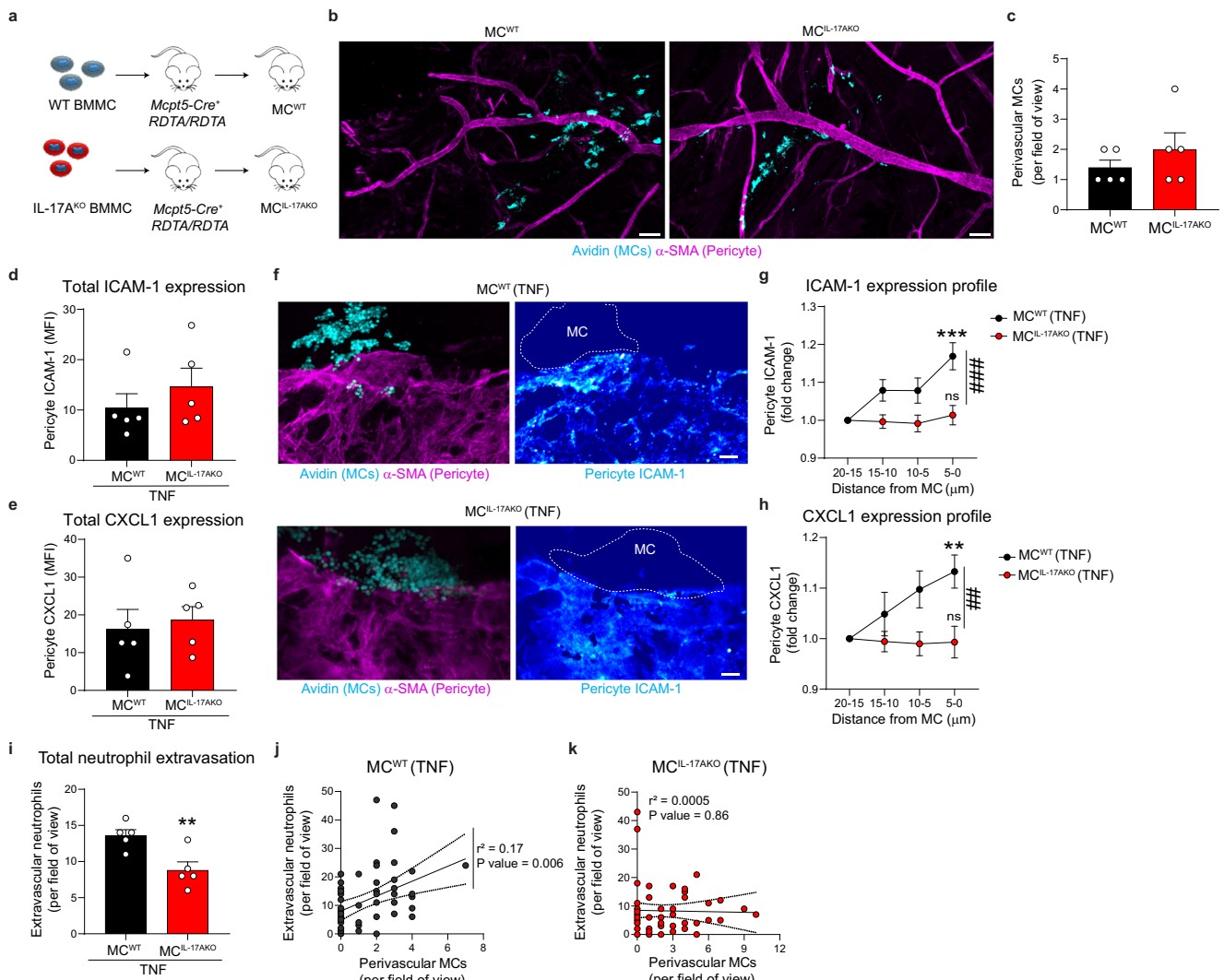

**Fig. 6 | MC-derived IL-17A induces graded expression of ICAM-1 and CXCL1 in pericytes.** MC<sup>deficient</sup> animals were injected with BMMC from WT (MC<sup>WT</sup>) or IL-17A<sup>KO</sup> (MC<sup>IL-17AKO</sup>) mice. Four months later, MC-reconstituted mice were stimulated with TNF (300 ng) for 4 h. Cremaster muscles were collected and immunostained for neutrophils (MRP14), MCs (avidin), pericytes (α-SMA) and ICAM-1 or CXCL1. **a** Generation of MC-reconstituted mice. **b** Low-magnification image of the cremasteric microcirculation in MC<sup>WT</sup> and MC<sup>IL-17AKO</sup> mice depicting the reconstitution of MCs, scale bars 100 μm. **c** Number of perivascular MCs in MC<sup>WT</sup> and MC<sup>IL-17AKO</sup> mice (n = 5 mice). **d**, **e** Quantification of **d** ICAM-1 and **e** CXCL1 MFI on pericytes (n = 5 mice). **f** Images of post-capillary venules in TNF-treated MC<sup>WT</sup> and MC<sup>IL-17AKO</sup> mice showing increased expression of ICAM-1 in MC<sup>WT</sup> but not in MC<sup>IL-17AKO</sup> mice on pericytes, scale bars 5 μm. **g** Quantification of pericyte ICAM-1 MFI in regions at 5 μm interval from a perivascular MC in TNF-treated MC<sup>WT</sup> and MC<sup>IL-17AKO</sup> mice (n = 28 MC<sup>WT</sup>, 27 MC<sup>IL-17AKO</sup> perivascular MC regions, data pooled from five mice, ***p

value < 0.0001, ###p value < 0.0001). **h** Quantification of pericyte CXCL1 MFI in regions at 5 μm interval from a perivascular MC in TNF-treated MC<sup>WT</sup> and MC<sup>IL-17AKO</sup> mice, (n = 19 MC<sup>WT</sup>, 27 MC<sup>IL-17AKO</sup> perivascular MC regions, data pooled from five mice, **p value = 0.0088, ##p value = 0.0012). ICAM-1 and CXCL1 MFI were normalised to the more distal region (i.e., 15–20 μm). **i** Extravascular neutrophils in MC<sup>WT</sup> or MC<sup>IL-17AKO</sup> mice (n = 5 mice, p value = 0.0094). **j**–**k** Correlation of the number of extravascular neutrophils and perivascular MCs in **j** MC<sup>WT</sup> (n = 58 venules) and **k** MC<sup>IL-17AKO</sup> (n = 60 venules); data pooled from five mice. Line indicated linear regression and dashed lines 95% confidence band. Mean±SEM (each mouse represents one independent experiment). **c**–**e**, **i** two-tailed Student's t test; **g**, **h** two-way ANOVA followed by Sidak's post-hoc test; **j**, **k** Spearman's rank correlation test. **p < 0.01, ***p < 0.001 as compared to MC<sup>WT</sup> or 20-15 μm region or as indicated ##p < 0.01, ###p < 0.001 (ns = not significant). Source data are provided as a Source Data file.

potential means of enhancing neutrophil effector functions in the interstitial tissue[11,57]. Moreover, our study provides further evidence for the crucial role of functional gradients during immune cell trafficking and their establishment by the composition of the local environment. In conclusion, the molecular and cellular pathway described here provide novel insights into the mechanisms through which anti-inflammatory effects of anti-IL-17A therapy are achieved in the clinic, especially in the context of TNF-driven inflammatory pathologies. Furthermore, targeting the IL17A-MC-pericyte axis may offer new avenues for controlling deleterious inflammatory responses in pathologies associated with exuberant neutrophil and/or MC effector functions[58].

# Methods

## Antibodies

Anti-CD31 (390), mAbs from ThermoFisher; PB- & BV711-anti-mouse CD45 (30-F11 dilution 1/500), AF488-anti-CD115 (AFS98 dilution 1/300), PE-Cy7-anti-CD31 (390 dilution 1/300), AF488- & AF647-anti-CD54 (YN1/1.7.4 dilution 1/300), APC-anti-CD140b (18A2 dilution 1/300), APC-Cy7-anti-CD115 (AFS98 dilution 1/300), AF647- & PE-Cy7-anti-CD117 (2B8 dilution 1/300), PB-FcεRI (MAR1 dilution 1/300), AF647-anti-IL-17A (TC11-18H10.1 dilution 1/300), AF594-anti-CD4 (GK1.5 dilution 1/300), AF700-anti-CD3 (17A2 dilution 1/300), BV605-anti-CD41 (MWReg30 dilution 1/300), PE-anti-CD49d (R1-2 dilution 1/300), AF647-anti-CD11c (N418 dilution 1/300), Rat IgG1 mAbs from

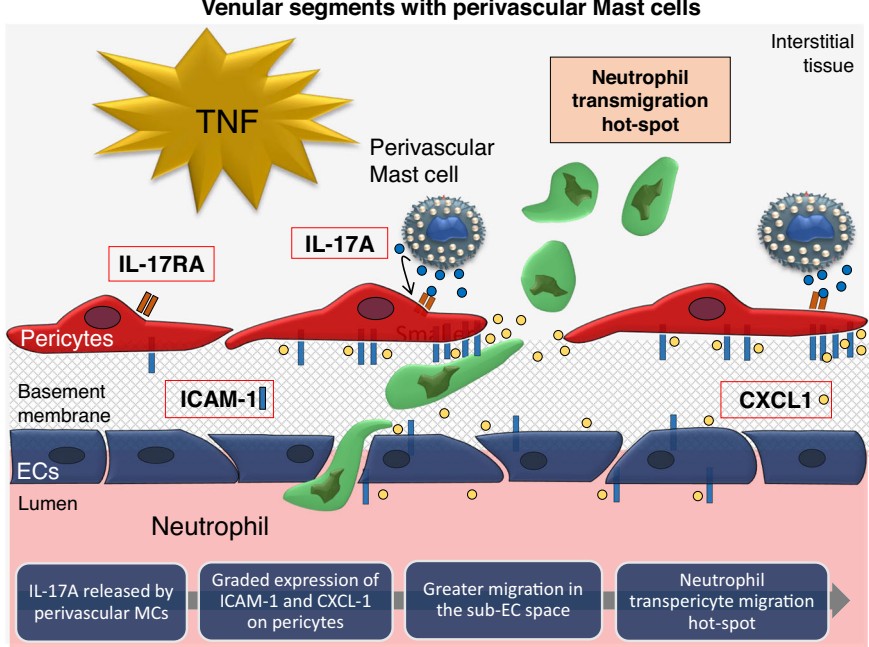

**Fig. 7 | A Model for the role of perivascular mast cells in guiding neutrophils out of the pericyte layer of blood vessels.** Upon acute inflammation, perivascular MCs release the cytokine IL-17A that promotes localised enrichment of intercellular adhesion molecule 1 (ICAM-1) and chemokine CXCL1 on nearby pericytes from post-capillary venules. This response induces a graded expression of those the key molecules within the pericyte layer, allowing for the directed migration of neutrophils in the subendothelial space and subsequent exit from the vessel wall towards the nearest MCs.

Biolegend; Anti-IL-17RA (G-9 dilution 1/100) from SantaCruz; Desmin (D33 dilution 1/100) were obtained from Dako; Anti- αSMA (1A4 dilution 1/300) from Sigma-Aldrich; anti-mouse CXCL1 (polyclonal dilution 1/100) from R&D systems. Anti-mMCP-1 (RF6.1 dilution 1/100) from ThermoFisher. Anti-MRP14 mAb (dilution 1/500) was a gift form Dr Nancy Hogg (The Francis Crick Institute, UK).

### Animals

Male WT C57BL/6 (stock number JAX #000664, Charles River, UK), *LysM-EGFP-ki (Gift from Dr M. Sperandio (Ludwig Maximilians University Munich, Germany)*[59], *α-SMA-RFPcherry-Tg (Gift from Dr D. Rowe (University of Connecticut Health Center, US), LysM-EGFP-ki; α-SMA-RFPcherry-Tg* mice (8-12 weeks old) were used for all studies. *Mcpt5-Cre-ROSA26-YFP* and *Mcpt5-Cre-RDTA/RDTA* were provided by A. Roers (Institute for Immunology, Heidelberg University Hospital, Heidelberg, Germany) and generated as previously described[21]. *Mcpt5-Cre-RDTA/ RDTA* were crossed with *LysM-EGFP-ki; α-SMA-RFPcherry-Tg* animals to generate *Mcpt5-Cre-RDTA;LysM-EGFP-ki; α-SMA-RFPcherry-Tg. Mcpt5-Cre-RDTA;LysM-EGFP-ki; α-SMA-RFPcherry-Tg* expressing the Cre-recombinase were referred as MC^deficient, littermates *Mcpt5-Cre⁺-RDTA;-LysM-EGFP-ki; α-SMA-RFPcherry-Tg* (MC^ctrl) were used as controls. Of note, no difference in level of MC deficiency was observed between *Mcpt5-Cre⁺-RDTA/RDTA* (0.0 ± 0.06) and *Mcpt5-Cre⁺-RDTA/-* mice (0.0 ± 0.00). Il17^atm1.1(icre)Stck (stock number JAX #016879) mice, referred as "IL-17A^KO", were purchased from Jackson Laboratory (Maine, US) and generated as previously described[60]. In these animals the endogenous *Il17a* gene has been substituted with a Cre-recombinase gene insert inducing total IL-17A deficiency in homozygous mice.

MC reconstitution was performed as previously described[25]. Briefly, BMMCs were derived from WT or Il17^atm1.1(icre)Stck mice. Bone marrow cells were isolated from the femur of animals. After 3 weeks of differentiation in presence of 10 ng/ml of interleukin 3, mature BMMCs, validated by flow cytometry for high expression of FcεRI and CD117, were injected i.v. (10⁶) and locally in the scrotal (i.s.) cavity (10⁶). Four months post-engraftment, mice subjected to TNF-stimulation were analysed for neutrophil infiltration, ICAM-1 and CXCL1

expression. No tissue-infiltrated neutrophils were observed in mice injected with BMMCs at steady-state. BMMCs from WT or Il17^atm1.1(icre)Stck mice showed similar level of purity (~98%) and expression level of FcεRI and CD117. All animals were group housed in individually ventilated cages (maximum of 5 mice per cage) under specific pathogen-free (SPF) conditions and a 12-hour (h) light-dark cycle. Room temperature and humidity were maintained within 18-20 °C and 30-70% humidity. Food and water were provided ad libitum. At the end of the experiments, mice were euthanised using cervical dislocation. All in vivo experiments were conducted at the William Harvey Research Institute, Queen Mary University of London, UK under the UK legislation for animal experimentation (UK Home Office licence number PPL: P873F4263) and in agreement with the UK Home Office Animals Scientific Procedures Act 1986 (ASPA).

### Confocal IVM

Anaesthetised (isofluorane, 3%) male mice received an (i.s.) injection of fluorescently labelled anti-CD31 mAb (4 μg), anti-CD117 mAb (10 μg) and TNF (300 ng, R&D Systems) in a 400 μl bolus to label vessels and MCs within the tissue and induce an acute inflammatory response, respectively. Control animals received PBS. Of note, tissue macrophages, exhibiting dim levels of GFP expression and phagocytosing the anti-CD31 and CD117 mAbs, were excluded from the analysis. Injections of anti-CD31 and CD117 mAbs by themselves did not induce neutrophil recruitment. The cremaster muscles were then prepared for intravital imaging 2 h post TNF administration, respectively, as described[19,61]. In some experiments, anti-mouse IL-17A mAb or control IgG1 (50 μg) were injected i.s. with TNF as indicated in relevant texts. Z-stack images of post-capillary venules (20-40 μm in diameter) were captured using Leica SP8 confocal microscope (LEICA LAS-X) incorporating a x20 water-dipping objective (NA 1.0), as detailed previously[9,19].

### Quantification of neutrophil TEM and TPM

Still images and 4D live recordings were analysed using IMARIS software™ (Bitplane). Normal neutrophil TEM was classified as a response

in which the cells migrated through EC junctions in a luminal-to-abluminal direction, as previously described[19,61]. Neutrophil abluminal crawling parameters were quantified as previously described[9,10]. Neutrophil TPM was defined as cells migrating through the pericyte layer in an abluminal-to-interstitial direction. Neutrophil TPM hotspots were defined as an area of 16 mm$^2$ in the pericyte sheath where at least three neutrophils entered the interstitium during the whole duration of the recording (i.e., 2 h). Neutrophil migration dynamics (speed, length, duration, straightness and displacement) were determined by manual tracking of individual neutrophils using Imaris software as previously described[9].

### Bright-field IVM

Mice were injected i.s. with TNF (300 ng) or PBS alone for 2 h prior to cremaster muscle exteriorisation. Leukocyte rolling and firm arrest within 20–40 μm post-capillary venules were quantified by IVM over a 2 h period using a bright-light microscope within multiple vessel segments (3–5) of several vessels (3–5) per mouse (Axioskop FS, Carl Zeiss, UK), as previously detailed[62]. Several vessel segments (3–5) from multiple vessels (3–5) were quantified for each animal.

### Immunofluorescence staining and confocal analysis

Whole-mount cremaster muscles, ear skin, heart or hindlimb (calf) muscle sections were analysed for visualisation of neutrophils (anti-MRP14 mAb), MCs (avidin), pericytes (α-SMA), ECs (CD31), ICAM-1 and CXCL1 as previously published[9,61]. Briefly, following stimulation TNF (300 ng, R&D Systems) or LPS (300 ng, Sigma), surgically removed tissues were fixed in ice-cold PFA (4% in PBS) for 45 min, blocked and permeabilised at room temperature for 4 h in PBS containing 25% FCS and 0.5% Triton X-100, and incubated overnight at 4 °C with primary antibodies. Ear skin layers were separated before the staining. ICAM-1 labelling was achieved by i.s. injection of 5 μg anti-ICAM-1-AF647 20 min before the end of the reaction. Immunostained tissues were imaged with either a Leica SP8 or a Zeiss 800 confocal laser-scanning microscope. Serial Z-stacks of post capillary venules were acquired using a water-dipping ×20 (1 NA) objective, oil immersion ×40 (1.3 NA) or ×63 (1.4 NA) objectives at a resolution of 1024 × 512× -40 pixels. To capture fields up to 3 mm × 3 mm of tissues, tile scan acquisition was performed where necessary using 10% tile overlap; tiles were stitched and fused using the ZEN software v2.6 (Zeiss, Germany).

Tissue-resident MC, macrophages and dendritic cell numbers and neutrophil extravasation responses in cremaster muscles were determined by staining using avidin and mAbs against MRP14 for MCs and neutrophils, respectively, as previously detailed[9,63]. Perivascular tissue-resident leukocytes were located less than 20 μm away from the venular wall. For analysis of ICAM-1 and CXCL1 expression, isosurfaces were created from specific channels delineating ECs, pericyte and MC based on regions immunostained for CD31 (CD31$^{high}$ junctional and CD31$^{dim}$ non-junctional regions), α-SMA and avidin, respectively. Deconvolution analysis was performed using Huygens software (SVI). Deconvolution parameters were the following: iterations numbers were set at 400,000, signal-to-noise ratio at 3 and quality threshold at 0.01.

All protein expression levels were quantified from 6 to 12 images/tissue and expressed as MFI values of tissues stained with specific antibodies post-subtraction of MFI values acquired from tissues stained with isotype control antibodies. Line intensity measurements were performed using Fiji/ImageJ. ICAM-1 and CXCL1 graded expression were analysed using isosurfaces created based on α-SMA staining and distance from a perivascular MC. For each ICAM-1 and CXCL1 enriched regions, MFI of the isosurfaces placed every 5 μm from a perivascular MCs were normalised to the MFI of the most distant isosurface (i.e., 15–20 μm) and expressed as fold change.

### Quantification of inflammatory mediators

Anaesthetised (isofluorane 3%) mice were injected i.s. with TNF (300 ng, R&D Systems) in 400 μl PBS. Control animals received PBS only. Cremaster muscles were harvested 4 h later. Tissues were homogenised in PBS containing 0.1% Triton X-100 and 1% Halt Protease and Phosphatase Inhibitor cocktail (ThermoFisher) and mechanically dissociated using the Precellys24 beat-beading system (Bertin Technologies). Levels of IL-17A & CXCL1 were analysed as per the manufacturer's instructions by ELISA (R&D Systems, sensitivity: 16 and 2 pg/ml respectively). The quantity of mediators detected in tissues was normalised to protein content as determined using a BCA assay (Thermo Fisher).

### Flow cytometry

Whole blood was collected through the hepatic vein in PBS + 50 mM EDTA. Indicated organs were harvested, mechanically dissociated and treated with 625 U/mL Collagenase I (ThermoFisher) and 100 U/ml DNAse I (Sigma-Aldrich) for 30 min at 37 °C. Where required, samples were treated with ACK buffer (150 mM NH3Cl, 1 mM KHCO3 and 1 mM EDTA) to lyse red blood cells. Subsequently, single-cell suspensions were incubated with anti-CD16/-CD32 antibodies (Becton Dickinson) to block Fc-receptors and stained with primary fluorescently labelled antibodies of interest. Dead cells were excluded using Zombie Aqua (Biolegend) (Fig. S5). The samples were then analysed using an LSR Fortessa flow cytometer (Becton Dickinson) and FlowJo software (TreeStar).

### Analysis of IL-17A mRNA expression

MCs were isolated from cremaster muscles (CD45$^+$, FcεRI$^+$, CD117$^+$) and RNA extraction was performed using a RNeasy Mini kit (Qiagen) as per the manufacturer's instructions. Gut lamina propria (LP) tissue enriched in IL-17A producing cells and retinal pigment epithelium (RPE) with low numbers of IL-17A producing cells at steady state[64] were used as controls. Reverse transcription was performed using the iScriptTM cDNA synthesis kit (Biorad) and amplicons generated by PCR using GoTaq polymerase (Promega) using standard procedures. IL-17A primer sequences were as follows: sense: ATCCCT-CAAAGCTCAGCGTGTC and anti-sense: GGGTCTTCATTGCGG TGGAGAG. GAPDH primer sequences were as follows: sense: TCGTGGATCTGACGTGCCGCCTG and anti-sense: CACCACCCT GTTGCTGTAGCCGTAT.

### Cell isolation and culture

Cremaster muscles from α-SMA-RFPcherry-Tg or WT mice were digested with 625 U/mL Collagenase I (ThermoFisher) and 100 U/mL DNase I (Sigma-Aldrich) in PBS for 45 min at 37 °C. For pericyte isolation, the resulting cell suspension was seeded onto tissue culture plates coated with 0.1% gelatin and collagen I (Advanced BioMatrix) and cultured in low glucose Dulbecco's Modified Eagle's Medium (DMEM) supplemented with 10% FCS, 100 U/mL penicillin, 100 mg/mL streptomycin (ThermoFisher) and 100 pM pigment epithelium-derived factor (PEDF, Sigma-Aldrich). After 21 days of culture, confluent cells were detached with 5 mM EDTA and cells exhibiting the unique venular pericyte signature (α-SMA$^+$PDGFR-β$^+$NG2$^-$) were isolated using the FACSAria cell sorter (Becton Dickinson). α-SMA$^+$ cells were identified by RFP expression. Pericytes showing >90% purity were subjected to further analyses. Mast cells were cultured in OPTI-MEM supplemented with 10% FCS, 100 U/mL penicillin, 100 mg/mL streptomycin (Thermo-Fisher) and 4% CHO transfectants secreting murine SCF (a gift from Dr P. Dubreuil, Marseille, France, 4% correspond to -50 ng/ml SCF) for 3 weeks[65]. Mast cell purity was assessed by flow cytometry (CD117$^+$/FcεRI$^+$). Blood neutrophils were negatively sorted using the EasySep™ Mouse Neutrophil Enrichment Kit (StemCell) according to manufacturer instructions.

## Transwell chemotaxis assay

Pericytes (~10,000) were seeded on the top well of a ChemoTx® Disposable Chemotaxis System (3 μm pore diameter, Neuroprobe) for 48 hours. Cremaster-derived MCs (~5000) were placed in the bottom chamber in Tyrode's buffer (Sigma) supplemented with 0.5% BSA. Cells were stimulated with TNF (10 ng/ml) in the bottom chamber and immediately ~30,000 neutrophils were placed on the top chamber for 1 hour. In one condition, a blocking anti-IL-17A mAb (10 μg/ml) was added in parallel with TNF stimulation. Neutrophils migrated into the bottom chambers were resuspended in PBS containing 5 mM EDTA and their absolute numbers were determined by flow cytometry.

**Statistics and reproducibility.** Sample size is indicated in the figure legend for each experiment. The level of significance was set at 5%, and the power was set at 80%. For cell-based quantitative experiments, results of multiple independent biological replicates were used (at least 3). Mice used in the present study were randomly assigned to each group and no data were excluded from the analyses. The researcher performing the animal experiment was blinded for the different animal groups when technically possible. Mouse stimulation and data collection were performed by different individuals. Data was decoded after analysis. Data analysis was performed using GraphPad Prism v8 & v9 (GraphPad software). Results are expressed as mean ± SEM and the n numbers for each dataset are provided in the figure legends. Statistical significance was assessed by two-tailed Student's $t$ test, one-way followed by Tukey's post-hoc test or two-way ANOVA followed by Tukey's or Sidak's post-hoc test. $P$ values <0.05 were considered significant.

## Study approval

All in vivo experiments were conducted under UK legislation according to the Animal Scientific Procedures Act 1986, with all procedures being conducted in accordance with UK Home Office regulations.

## Reporting summary

Further information on research design is available in the Nature Portfolio Reporting Summary linked to this article.

## Data availability

Raw confocal image files are stored on servers at William Harvey Research Institute, Queen Mary University of London due to their large size. All raw data from the study are available from the corresponding author upon request. Source data are provided with this paper.

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

## Acknowledgements

This work was supported by funds from the Wellcome Trust (098291/Z/12/Z to S.N.) and British Heart Foundation (PG/17/85/33395 to R.J., M-B.V. and S.N., PG/19/73/34663 to M-B V. and J.A.C.). R.J. is supported by fellowships from the "Fondation pour la Recherche Medicale FRM" (award SPE20170336775), Fondation Bettencourt Schueller (Prix Jeunes Chercheurs 2017), the People Programme (Marie Curie Actions) of the EU's 7th Framework Programme (FP7/2007-2013) under REA grant agreement no. 608765 and a BHF CRE Research Fellowship from Imperial college (RE/18/4/34215). IMGF was supported by the Newton Advanced Fellowship from the Royal Society (NAF/R1/180017 awarded to MS and SN). The authors would like to thank Dr Anna Barkaway for critical reading of the manuscript. This work was supported by the CMR Advanced Bio-Imaging Facility, which has been established through generous funds from the Wellcome Trust, the British Heart Foundation, Barts Charity and QMUL.

## Author contributions

R.J., M-B.V. and S.N. designed the experiments; R.J. did most of the experiments, compiled and analysed the data. I.M.G-F, T.G., J.A.C., L.V.-M., E.L., M.S., J.W. contributed to certain key experiments. R.J. prepared the figures; M.S, D.V. and A.R. provided resources and advised on experimental protocols; R.J., S.N. and M-B.V. wrote the manuscript; R.J., M-B.V. and S.N. funded the work.

## Competing interests

The authors declare no competing interests.
