## [Peer Review File · Nature Communications]

Neutrophil breaching of the blood vessel pericyte layer during diapedesis requires mast cell-derived IL-17AREVIEWER COMMENTS

Reviewer #1 (Remarks to the Author):

The manuscript entitled "Mast cell-derived IL-17A facilitates neutrophil diapedesis by mediating neutrophil breaching of the pericyte layer in vivo" by Joulia et al reported that perivascular mast cells regulate neutrophil migratory pattern within the sub-endothelial space of the inflamed venules in mast cell-derived IL-17A dependent manner. The authors mainly employed confocal intravital imaging of the exteriorized mice cremaster to verify the conclusion of this study under control vs blockade of IL-17A conditions.

IL-17A are produced from various kinds of cells including neutrophil itself and also have roles in driving inflammatory events in other immune cells such as Th17 cells. As shown in referred papers of this manuscript (12, 13, 14), mast cells were evenly distributed close to most microvessels in the imaging region from both whole-mount labeling and two-photon intravital imaging of the mice ear skin. Although the authors performed intravital imaging of the cremaster and whole-mount labeling of cremaster muscle/ear skin, the reviewer do not find the rationale or evidence for uneven distribution of perivascular mast cells close to specific microvessels.

A reference (13) in this manuscript showed that MC-derived TNF induced neutrophil intraluminal crawling by infusion into the bloodstream, although the disease model was different. If MC-derived TNF could be infused into the bloodstream, it would be also possible for MC-derived IL-17A to be infused into the bloodstream. To validate novel finding that the role of MC-derived IL-17A in sub-endothelial behavior in the sequential steps of neutrophil extravasation, the author should verify how distinct proinflammatory mediators from the same resident mast cells can spatiotemporally affect different steps of neutrophil extravasation cascade.

As neutrophils also produce autocrine IL-17A, the author should also consider the distinct role of autocrine IL-17A from MC-derived IL-17A for neutrophil migration step (sub-endothelial movement in this study).

<https://pubmed.ncbi.nlm.nih.gov/24362892/>

Major novel finding in this manuscript is based on the authors' quantification and interpretation from immunostaining and intravital imaging of the cremaster muscle (and ear skin) in Fig. 1a, 1g, 1i, 2c, 3d, 3g, and more with or without supplementary movies. Supplemental movies should be critical evidence to support the authors' conclusion. However, all of supplementary movies in this manuscript were immediately zooming in smaller region to visualize a migrating cell or small number of migrating cells after rotation of the images. The migration of a few cells in the smaller region of the whole movie size never support the authors' conclusion. Data in the manuscript for the authors' conclusion is not enough to publish in this journal.

Reviewer #2 (Remarks to the Author):

the authors showed the role of mast cells produce IL-17A after TNF stimulation. IL-17A acts on pericytes and mediates breach allowing the diapedesis on neutrophils. Data are new, convincing and show the significant role of mast cells as key regulators in TNF inflammatory process. authors prevent few questions by showing some negative results in their process to demonstrate functional activities.

i may have few minor comments. An experiment with a full depletion of mast cells using the Wsh mice or other KI mice allowing depletion of mast cells could confirm their results. Even if macrophages don't show an upregulation of IL17, it could be interesting to see where the macrophages are localized and may respond for IL17 secretion in a coculture with mast cells after TNF stimulation. What about the DC localized around?

Pericyte culture is very difficult, so if this may be done, it could be interesting to perform a transwell experiment with pericytes in the upper level, TNF activated mast cell in the lower well and check neutrophil migration.

In an experiment in Wsh mice (or any other Mast cell-depleted mouse model) stimulated with TNF, do the authors can observe an upregulation of ICAM1, CXCL1 on pericytes. and with IL17 injection?

basophile recruitment and basophile IL17 production could be excluded?

Are these results on pericytes could be detected after mast cell degranulation (IgE or LPS stimulation)

Response to Reviewers' Comments:

Reviewer #1:

We are grateful for the reviewer's detailed analysis of our work. Acting on them has substantially improved our manuscript.

A) *Diversity of leukocytes expressing IL-17A.*

The Reviewer is absolutely correct in stating that many leukocytes produce IL-17A. In the original submission we demonstrated the expression of IL-17A by tissue resident mast cells and macrophages (Fig. 3b-c and Supplementary Fig. 4a-b), however all our data indicate that only mast cell derived IL-17A (as demonstrated with the use of MC deficient and chimeric animals) induced the regulated patterning of CXCL1 and ICAM-1 expression by pericytes and the occurrence of neutrophil hotspots in the vicinity of perivascular MCs (Fig. 1i-j, Fig. 2f-g, Fig. 5, Fig. 6). To provide further support for this data, we have now performed additional experiments showing that during acute neutrophil recruitment following TNF-induced inflammation (4 hrs), T helper cells (and hence Th17) and basophils are not detected in inflamed tissues (new Supplementary Fig. 4e-f). This new data excludes their contribution in IL-17A production in our TNF-driven model. Furthermore, flow cytometry analyses showed that tissue infiltrated neutrophils do not express IL-17A (new Supplementary Fig. 4d), ruling out the potential existence of an autocrine effect of neutrophil-derived IL-17A. Whilst we employed a sterile inflammation to induce neutrophil recruitment, IL-17A production by neutrophils has been reported following microbial infections or chronic inflammatory disorders, although these findings are contentious (Huppler, Verma et al. 2015, Tamassia, Arruda-Silva et al. 2018).

The new data are incorporated in new Supplementary Fig. 4d-f and are discussed in the main text (results and discussion).

B) *Rationale for uneven distribution of perivascular mast cells close to specific microvessels.*

This is an interesting comment. Our data indeed demonstrate a heterogenous distribution of MCs around blood vessels of murine cremaster muscles. Further detailed analyses of their distribution in the cremaster muscle indicated that perivascular MCs are preferentially located around arterioles, postcapillary venules (PVCs, in particular around vessels with a diameter of 20 to 40 μm) but not capillaries (new Supplementary Fig. 1b-c).

Moreover, we have now compared the localisation and distribution of MCs in the cremaster muscle with other organs such as the ear skin and two non-mucosal internal organs: the heart and leg muscle. We observed that MCs were heterogeneously distributed around blood vessels from the myocardium and the leg muscle (new Supplementary Fig. 1a). In contrast, in the ear skin, the density of MCs was higher and more evenly distributed, as supported by previous studies cited in our MS (e.g. References 12-15 and 35). These data may indicate differential recruitment and patterning of MC progenitors in different tissues.

Fundamentally, despite the different expression profile our data clearly shows that neutrophil recruitment to inflamed tissues (e.g. cremaster muscle and skin) is correlated to the number of perivascular mast cells (Fig.1 b-c), irrespective of their density or distribution profile. The mechanisms of the distribution of perivascular MCs in different tissues is not fully understood but a previous study

from Christer Betscholtz's lab (He, Vanlandewijck et al. 2018) showed that pericytes express Stem Cell Factor (SCF/Kitl) (<https://betsholtzlab.org/VascularSingleCells/database.html>), a potent chemoattractant and main development factor for MCs (Nilsson, Butterfield et al. 1994, Yee, Paek et al. 1994). Studies by Cheng et al. and Dudeck et al. have identified MCs in close association with different types of blood vessels (Cheng, Hartmann et al. 2013, Dudeck, Kotrba et al. 2021). Furthermore, a newly preprint study from Tim Lammermann's lab demonstrates that in vivo, MCs are anchored leukocytes exhibiting minimal motility in tissues and that their distribution along arterioles (but not venules or capillaries) is controlled by β 1-integrin and the heterogeneous expression of KIT ligands by stroma cells (including ECs and VSMC) (Bambach, Kaltenbach et al. 2022). Collectively, whilst elucidating the distribution of MCs is beyond the scope of this study that focuses on their implication for neutrophil-pericyte interaction in vivo, we have amended our discussion to account for these interesting observations.

C) *Diffusion of MC-derived IL-17A in the blood stream and potential impact on neutrophil extravasation steps.*

As detailed in the original manuscript, we have found that MCs and IL-17A impact neutrophil-pericyte interactions, but not neutrophil-endothelial interactions, in vivo. Specifically, using MC deficient animals or IL-17A blockade/deficiency we did not observe changes in neutrophil rolling, adhesion or trans-endothelial migration as compared to control animals (ie WT littermates or vehicle treated mice) (Fig. 2b and Supplementary Fig. 3e-f). These observations are supported by the literature showing that in vitro, IL-17A facilitates the production of pro-inflammatory cytokines by human pericytes but not endothelial cells (Liu, Lauridsen et al. 2016). Furthermore, the latter study clearly demonstrates that IL-17A stimulated pericytes, but not ECs, promotes neutrophil polarisation, survival and activation. Hence, our data, in conjunction with previously published works collectively suggest that MCs and MC-derived IL-17A are not implicated in the first steps of the extravasation cascade.

The study by Dudeck et al. reports that within a murine skin hypersensitivity inflammatory model, MC-derived TNF can diffuse into the blood stream and via this mechanism impact neutrophil diapedesis (ref 13). To investigate the potential diffusion of MC-derived IL-17A into the blood circulation, we have specifically quantified the presence of IL-17A in plasma at steady state (PBS) and in TNF-stimulated tissues of both WT and MC deficient animals. Our new data categorically shows that in our acute inflammatory model, no IL-17A is detectable in the plasma of WT or MC deficient animals (new Supplementary Fig. 4g).

D) *Better exemplification of the occurrence of neutrophil extravasation in the vicinity of perivascular MCs. More evidence from video recordings to support authors' conclusions*

As advised, we have now added 6 new supplementary videos at different magnifications, showing multiple sites of neutrophil TEM and TPM in relation to localisation of perivascular MCs for each of our experimental conditions (new Supplementary videos 1, 2, 4, 6, 8 & 9).

Reviewer #2:

We are delighted the Reviewer considers our work to be new and convincing and has only raised minor points. These have been addressed below with additional experiments.

A) *Using other MC deficiency models*

Here we used the Mcpt5-Cre-RDTA/RDTA mice that is characterized by the genetic deletion of connective-tissue MCs. Using detailed flow cytometry analysis, we have shown all MCs (FcεRI⁺/CD117⁺) from mouse cremaster muscles are positive for avidin signal (Supplementary Fig. 4b), suggesting that all MC granules express heparin, a specific characteristic of connective tissue MCs.

To further confirm this observation, we have now analysed the expression of mMCP-1 (Mcpt1) on mast cells, a marker for mucosal MCs. Our data clearly indicate that no MCs present in the cremaster muscle are positive for mMCP-1 (new Supplementary Fig. 3a).

Furthermore, flow cytometry analyses of tissues derived from our MC^{deficient} (Mcpt5-Cre⁺-RDTA/RDTA) animals showed a total absence of all type of MC (FcεRI⁺/CD117⁺; new Supplementary Fig. 3d), supporting the use of our GM mouse line in our study.

Finally, whilst other pan-MC deficient lines may be useful, several studies have reported on MC-independent inflammatory effects in c-KIT KO mice that is likely due to the importance of c-KIT and its ligand for the haematopoietic niche (Feyerabend, Weiser et al. 2011, Gutierrez, Muralidhar et al. 2015).

As such we believe that there are no ethical or scientific justifications for the use of another animal model of MC deficiency.

B) *Macrophage perivascular distribution and their role in IL-17A- dependent regulation of pericyte function in regulating neutrophil migration.*

We thank the Reviewer for this insightful comment which we have now addressed experimentally. Macrophages are present in the cremaster muscle and show a dense and homogenous distribution within the tissue, i.e. perivascular and non-perivascular localisation (new Supplementary Fig. 1d, f and g). In contrast to MCs, perivascular macrophages could be detected around all type of vessels, i.e. capillaries, arterioles and post-capillary venules (new Supplementary Fig. 1d and f). However, in contrast to perivascular MCs, macrophages were not significantly aligned with a specific size of post-capillary venules (new Supplementary Fig. 1g). Of note, IL-17A transcript was detected in those tissue-resident cells (Supplementary Fig. 4a), whilst the protein intracellular levels were ~2/3 lower than those detected in MCs (Fig 3b). Most importantly, IL-17A expression by macrophages was not altered during TNF-induced inflammation.

We then investigated a potential impact of IL-17A on tissue resident macrophages. We first analysed by flow cytometry their surface expression of IL-17RA, the key receptor chain for IL-17A signalling (Gaffen 2009). No significant levels of IL-17RA could be detected on the surface of tissue resident macrophages in control or inflamed tissues (new Supplementary Fig. 6d). This observation is supported by the literature demonstrating that IL-17RA is differentially expressed by distinct subclasses of macrophages; i.e. mostly detected in monocyte-derived or gut and peritoneal macrophages but not expressed by the resident pool in non-mucosal tissues (Das Sarma, Ciric et al.

2009, Barin, Baldeviano et al. 2012, Ge, Hertel et al. 2014, Lages, Simmons et al. 2017). Collectively, these data indicate that mast-cell derived IL-17A may not affect macrophage functions in TNF-driven acute inflammatory conditions.

Of note, our original data (Fig. 6) showed that upon TNF-stimulation, the pericyte graded expression of ICAM-1 and CXCL1 near MCs was absent and the total neutrophil extravasation response was suppressed in chimeric animals exhibiting IL-17A genetic defect specifically in MCs but not in macrophages (WT macrophages); suggesting macrophage derived IL17A-derived is not important for those responses.

C) *Dendritic cell localisation*

The presence of DCs was investigated by immunofluorescence. We showed that CD11c⁺ dendritic cells were present all across unstimulated cremaster tissues with some cells exhibiting perivascular localisation around different types of blood vessels. Their number was slightly higher compared to perivascular MCs but we did not see any particular localisation in close proximity between DCs and MCs. These new data are incorporated in the new Supplementary Figure 1e-f and results section. Although this distribution is interesting, we think the interaction between DCs and MCs has been extensively studied and beyond the scope of our paper (Dudeck et al. J Exp Med 2017, Choi et al. Science 2018).

D) *Role of basophils in IL-17A production.*

We have analysed the presence of basophils in blood, control and inflamed cremaster muscles (see point F). This new data shows that whilst detected in the blood circulation, no significant number of basophils (<50 cells/tissue) could be detected in the cremaster muscles of naïve (PBS-treated) or TNF- or LPS-stimulated tissues, excluding their contribution to IL-17A production (new Supplementary Figure 4e).

E) *MC-pericyte cocultures to look at the effect of MCs and IL-17A for neutrophil migration in vitro*

As suggested, we performed a transwell in vitro assay in which primary pericytes isolated from cremaster muscle of WT mice were grown on the top chamber and mast cells derived from cremaster muscle of WT mice were placed in the bottom chamber with or without TNF. In some wells, IL-17A blocking monoclonal antibody was added to the bottom chamber. Primary neutrophils were then added on the top of the pericyte layer and 1 hour later, the number of migrated neutrophils into the bottom chamber was quantified. This new data (new Supplementary Figure 7) indicates upon TNF-stimulation, MCs induced the migration of neutrophils through the pericyte layer, a response that was inhibited in the presence of an IL-17A blocking antibody. Collectively these new data support our in vivo observations for a role for MC-derived IL-17A in neutrophil migration through the pericyte layer of blood vessels in vivo.

F) *Regulation of pericyte of ICAM1, CXCL1 in TNF and IL-17A -stimulated tissues of MC deficient animals*

We thank the reviewer for this comment. We have now quantified the expression of total ICAM-1 and CXCL1 on pericytes in MC^{deficient} and littermate control mice following stimulation with TNF or IL-17A in vivo. Our new data shows that following TNF injection, MC^{deficient} mice exhibit a significant increase in the expression of total ICAM-1 and CXCL1 as compared to PBS-treated animal. This response was moderately decreased as compared to their respective WT littermate controls (see new Supplementary Fig. 9).

In addition, we analysed the impact of the local injection of exogenous IL-17A on ICAM-1 expression by pericytes and neutrophil migration in WT animals. This new data shows that -17A on its own does not promote neutrophil recruitment, nor up-regulates the total expression of ICAM-1 by pericytes in vivo.

Figure 1: Locally injected IL-17A does not induce neutrophil infiltration or ICAM-1 up-regulation on pericytes in vivo. Cremaster muscles of WT mice were stimulated with IL-17A (50 ng) or PBS for 4h. Cremaster muscles were collected and immunostained for MCs (avidin), pericytes (α -SMA), neutrophils (MRP14) and ICAM-1. Extravascular neutrophils in PBS or IL-17A injected mice (n=4 mice). Quantification of ICAM-1 MFI (n=4-5 mice) on pericytes.

G) *MC-dependent regulation of pericyte responses in another inflammatory model.*

This is an important point and as such we have now investigated the role MCs and IL-17A for the graded expression of ICAM-1 and CXCL1 and neutrophil diapedesis in vivo following local administration of LPS (new Supplementary Fig. 10). Our new data indicate a ~60% reduction in total neutrophil infiltration in LPS-stimulated tissues from IL-17A^{KO} mice as compared to WT animals. Furthermore, detailed confocal microscopy analyses showed a graded expression of pericyte ICAM-1 and CXCL1 in close proximity to perivascular MCs (i.e. ~30% and ~60 increased signal for ICAM-1 and CXCL1 respectively) in LPS-stimulated WT mice but not in IL-17A deficient animals. Collectively, these new data indicate that the regulation of pericyte functions during neutrophil diapedesis by MCs and IL-17A is not restricted to TNF mediated stimulation but can be applied to other inflammatory models involving MC activation.

Reference list:

- Bambach, S. K., L. Kaltenbach, N. Aizarani, P. Martzloff, A. Gavrilov, K. M. Glaser, R. Thünauer, M. Mihlan, M. Stecher, A. Thiriou, S. Wienert, U. von Andrian, M. Schmidt-Suppran, C. Nerlov, F. Klauschen, A. Roers, M. Bajénoff, D. Grün and T. Lämmermann (2022). "Slow integrin-dependent migration organizes networks of tissue-resident mast cells." Biorxiv: 2022.2007.2019.500614.
- Barin, J. G., G. C. Baldeviano, M. V. Talor, L. Wu, S. Ong, F. Quader, P. Chen, D. Zheng, P. Caturegli, N. R. Rose and D. Cihakova (2012). "Macrophages participate in IL-17-mediated inflammation." Eur J Immunol **42**(3): 726-736.
- Cheng, L. E., K. Hartmann, A. Roers, M. F. Krummel and R. M. Locksley (2013). "Perivascular mast cells dynamically probe cutaneous blood vessels to capture immunoglobulin E." Immunity **38**(1): 166-175.
- Das Sarma, J., B. Ciric, R. Marek, S. Sadhukhan, M. L. Caruso, J. Shafagh, D. C. Fitzgerald, K. S. Shindler and A. Rostami (2009). "Functional interleukin-17 receptor A is expressed in central nervous system glia and upregulated in experimental autoimmune encephalomyelitis." J Neuroinflammation **6**(1): 14.
- Dudeck, J., J. Kotrba, R. Immler, A. Hoffmann, M. Voss, V. I. Alexaki, L. Morton, S. R. Jahn, K. Katsoulis-Dimitriou, S. Winzer, G. Kollias, T. Fischer, S. A. Nedospasov, I. R. Dunay, T. Chavakis, A. J. Müller, B. Schraven, M. Sperandio and A. Dudeck (2021). "Directional mast cell degranulation of tumor necrosis factor into blood vessels primes neutrophil extravasation." Immunity **54**(3): 468-483 e465.
- Feyerabend, T. B., A. Weiser, A. Tietz, M. Stassen, N. Harris, M. Kopf, P. Radermacher, P. Moller, C. Benoist, D. Mathis, H. J. Fehling and H. R. Rodewald (2011). "Cre-mediated cell ablation contests mast cell contribution in models of antibody- and T cell-mediated autoimmunity." Immunity **35**(5): 832-844.
- Gaffen, S. L. (2009). "Structure and signalling in the IL-17 receptor family." Nat Rev Immunol **9**(8): 556-567.
- Ge, S., B. Hertel, N. Susnik, S. Rong, A. M. Dittrich, R. Schmitt, H. Haller and S. von Vietinghoff (2014). "Interleukin 17 receptor A modulates monocyte subsets and macrophage generation in vivo." PLoS One **9**(1): e85461.
- Gurish, M. F. and K. F. Austen (2012). "Developmental origin and functional specialization of mast cell subsets." Immunity **37**(1): 25-33.
- Gutierrez, D. A., S. Muralidhar, T. B. Feyerabend, S. Herzig and H. R. Rodewald (2015). "Hematopoietic Kit Deficiency, rather than Lack of Mast Cells, Protects Mice from Obesity and Insulin Resistance." Cell Metab **21**(5): 678-691.
- He, L., M. Vanlandewijck, M. A. Mae, J. Andrae, K. Ando, F. Del Gaudio, K. Nahar, T. Lebouvier, B. Lavina, L. Gouveia, Y. Sun, E. Raschperger, A. Segerstolpe, J. Liu, S. Gustafsson, M. Rasanen, Y. Zarb, N. Mochizuki, A. Keller, U. Lendahl and C. Betsholtz (2018). "Single-cell RNA sequencing of mouse brain and lung vascular and vessel-associated cell types." Sci Data **5**: 180160.
- Huppler, A. R., A. H. Verma, H. R. Conti and S. L. Gaffen (2015). "Neutrophils Do Not Express IL-17A in the Context of Acute Oropharyngeal Candidiasis." Pathogens **4**(3): 559-572.
- Lages, C. S., J. Simmons, A. Maddox, K. Jones, R. Karns, R. Sheridan, S. K. Shanmukhappa, S. Mohanty, M. Kofron, P. Russo, Y. H. Wang, C. Chougnet and A. G. Miethke (2017). "The dendritic cell-T helper 17-macrophage axis controls cholangiocyte injury and disease progression in murine and human biliary atresia." Hepatology **65**(1): 174-188.
- Liu, R., H. M. Lauridsen, R. A. Amezcua, R. W. Pierce, D. Jane-Wit, C. Fang, A. S. Pellowe, N. C. Kirkiles-Smith, A. L. Gonzalez and J. S. Pober (2016). "IL-17 Promotes Neutrophil-Mediated Immunity by Activating Microvascular Pericytes and Not Endothelium." J Immunol **197**(6): 2400-2408.
- Nilsson, G., J. H. Butterfield, K. Nilsson and A. Siegbahn (1994). "Stem cell factor is a chemotactic factor for human mast cells." J Immunol **153**(8): 3717-3723.

- Tamassia, N., F. Arruda-Silva, F. Calzetti, S. Lonardi, S. Gasperini, E. Gardiman, F. Bianchetto-Aguilera, L. B. Gatta, G. Girolomoni, A. Mantovani, W. Vermi and M. A. Cassatella (2018). "A Reappraisal on the Potential Ability of Human Neutrophils to Express and Produce IL-17 Family Members In Vitro: Failure to Reproducibly Detect It." Front Immunol **9**: 795.
- Yee, N. S., I. Paek and P. Besmer (1994). "Role of kit-ligand in proliferation and suppression of apoptosis in mast cells: basis for radiosensitivity of white spotting and steel mutant mice." J Exp Med **179**(6): 1777-1787.

REVIEWERS' COMMENTS

Reviewer #1 (Remarks to the Author):

All issues raised from the original manuscript have been cleared in this revised manuscript. The reviewer believe that the manuscript is now acceptable for publication in this journal.

Reviewer #2 (Remarks to the Author):

The authors have well increased their results and have fully answered to all my previous comments and questions. In my opinion i accept this paper with these added result

Response to Reviewers' Comments:

We thank both reviewers for their suggestions to improve our manuscript and grateful they recommend the study to be published in Nature Communications.